# Neural responses in macaque prefrontal cortex are linked to strategic exploration

Caroline I. Jahn [1,2,3] *, Jan Grohn[1] *, Steven Cuell[1], Andrew Emberton[4], Sebastien Bouret[2], Mark E. Walton[1], Nils Kolling[5,6‡] *, Jérôme Sallet[1,6‡] *

1 Wellcome Centre for Integrative Neuroimaging, Department of Experimental Psychology, University of Oxford, Oxford, United Kingdom, 2 Motivation, Brain and Behavior Team, Institut du Cerveau et de la Moelle Epinière, Paris, France, 3 Sorbonne Paris Cité universités, Université Paris Descartes, Frontières du Vivant, Paris, France, 4 Biomedical Science Services, University of Oxford, Oxford, United Kingdom, 5 Wellcome Centre for Integrative Neuroimaging, OBHA, University of Oxford, Headington, United Kingdom, 6 Univ Lyon, Université Lyon 1, Inserm, Stem Cell and Brain Research Institute U1208, Bron, France

☯ These authors contributed equally to this work.
‡ NK and JS also contributed equally to this work.
* caroline.isabelle.jahn@gmail.com (CIJ); jan.grohn@psy.ox.ac.uk (JG); nils.kolling@inserm.fr (NK); jerome.sallet@inserm.fr (JS)

**Data Availability Statement:** The behavioral data, code to reproduce the figures shown in the manuscript and supplementary materials, and

## Abstract

Humans have been shown to strategically explore. They can identify situations in which gathering information about distant and uncertain options is beneficial for the future. Because primates rely on scarce resources when they forage, they are also thought to strategically explore, but whether they use the same strategies as humans and the neural bases of strategic exploration in monkeys are largely unknown. We designed a sequential choice task to investigate whether monkeys mobilize strategic exploration based on whether information can improve subsequent choice, but also to ask the novel question about whether monkeys adjust their exploratory choices based on the contingency between choice and information, by sometimes providing the counterfactual feedback about the unchosen option. We show that monkeys decreased their reliance on expected value when exploration could be beneficial, but this was not mediated by changes in the effect of uncertainty on choices. We found strategic exploratory signals in anterior and mid-cingulate cortex (ACC/MCC) and dorsolateral prefrontal cortex (dlPFC). This network was most active when a low value option was chosen, which suggests a role in counteracting expected value signals, when exploration away from value should to be considered. Such strategic exploration was abolished when the counterfactual feedback was available. Learning from counterfactual outcome was associated with the recruitment of a different circuit centered on the medial orbitofrontal cortex (OFC), where we showed that monkeys represent chosen and unchosen reward prediction errors. Overall, our study shows how ACC/MCC-dlPFC and OFC circuits together could support exploitation of available information to the fullest and drive behavior towards finding more information through exploration when it is beneficial.

statistical fMRI maps are available at: https://doi.org/10.5281/zenodo.7464572.

**Funding:** This research was supported by the Université Paris Descartes (doctoral and mobility grants to CIJ), the Medical Research Council UK (MR/K501256/1 and MR/N013468/1 to JG), St John's College, Oxford (JG), the Wellcome Trust (096587/Z/11/Z to SC, 090051/Z/09/Z and 202831/Z/16/Z to MEW, WT1005651MA to JS and the Wellcome Centre for Integrative Neuroimaging: 203139/Z/16/Z), the BBSRC (AFL Fellowship: BB/R01803/1 to NK), as well as the LabEx CORTEX of the Université de Lyon (ANR-11-LABX-0042 to JS). The funders had no role in study design, data collection and analysis, decision to publish, or preparation of the manuscript.

**Competing interests:** The authors have declared that no competing interests exist.

**Abbreviations:** ACC/MCC, anterior and midcingulate cortex; ANT, Advanced Normalization Tool; cOFC, central oribitofrontal cortex; dlPFC, dorsolateral prefrontal cortex; EPI, echo planar imaging; FEAT, fMRI Expert Analysis Toolbox; FLAME, FMRIB's Local Analysis of Mixed Effects; FSL, FMRIB Software Library; GLM, general linear model; GRE, gradient-refocused echo; ICI, inter-choice interval; lOFC, lateral orbitofrontal cortex; ITI, inter-trial interval; mOFC, medial orbitofrontal cortex; MRC, MRI-compatible screen; MrCat, Magnetic Resonance Comparative Anatomy Toolbox; OFC, orbitofrontal cortex; pgACC, pregenual anterior cingulate cortex; ROI, region of interest; vlPFC, ventrolateral prefrontal cortex; VOI, volume of interest.

# Introduction

In many species, most behaviors, including foraging, can be accounted for by simple behaviors—approach/avoidance of an observed and immediately available source of food—that require no mental representations. Exploration is, by definition, a non-value maximizing strategy [1,2], so in those models, exploration is often reduced to a random process, where noise in behavior can lead animals to change behavior by chance [1,3–6]. However, in species relying upon spatially and temporally scattered resources, such as fruits, exploration is thought to be aimed at gathering information about the environment in order to form a mental representation of the world. Work in monkeys and humans has shown that primates are sensitive to the novelty of an option when deciding to explore [7–9]. They sample novel options until they have formed a representation of their relative values compared to the available options. This work clearly showed that monkeys have a representation of the uncertainty and actively explore to reduce it. Similar results had been shown in humans, whose exploration is driven by the uncertainty about the options [10,11]. However, it is still unknown whether monkeys have a specific representation of potential future action and outcomes that enables them to organize their behavior over longer time or spatial scale. We design a novel paradigm in monkeys—based on work in humans—to assess whether on how monkeys engage in strategic exploration, which is exploring only when it is useful for the future. Strategic exploration enables an animal to adapt to a specific context and is essential to maximize rewards over a longer time and spatial scale. For frugivorous animals such as primates, it might be critical for survival.

Humans have been shown to strategically explore [12–15], but there is little evidence in other species. Inspired by Wilson and colleagues' "horizon" exploration task [12], we developed a task to investigate whether monkeys mobilize strategic exploration based on whether that information can improve subsequent choice. Importantly, non-human primate models provide insights into the evolutionary history of cognitive abilities, and of the neuronal architecture supporting them [16]. Given the rhesus monkeys' ecology (including feeding), they should also be able to use strategic exploration, but the extent to which they can mobilize strategic exploration might be different from that of humans. Based on the similarities in circuits supporting cognitive control and decision-making processes in humans and macaques [17,18], one could further hypothesize that the same neurocognitive processes (the same computational model) might be recruited but not to the same extent (different weights).

As in Wilson and colleagues' original study, we manipulated whether the information could be used for future choices by changing the choice horizon [12]. By comparing exploration in both conditions, we could test whether the animals reduced their reliance on value estimates (*random* exploration) and increased their preference for more uncertain options (*directed* exploration) when gathering information was useful for future choices in the long horizon [12]. In addition, we manipulated the contingency between the choice and the information by varying the type of feedback that monkeys received. In the complete feedback condition, information was freely available, and we could probe whether monkeys decreased their exploration compared to the classic partial feedback condition. In humans, providing the complete feedback decreases decision noise [19] and improves learning [20–23], both of which are consistent with reduced exploration. A strategic explorer would only actively explore—and forgo immediate rewards—when it is useful for the future (long horizon) and that it is the only way to obtain information (partial feedback). In addition to behavioral data, neural data were collected using fMRI to probe the neural substrates of strategic exploration. Our analysis was focused on regions previously identified in fMRI studies on reward valuation and cognitive control in monkeys [24–29]. Finally, we took advantage of the different feedback conditions to explore how monkeys update their expectations based on new information. Specifically, we

investigated the behavioral and neural consequences of feedback about the outcome of their choice and—in the complete feedback condition—on counterfactual feedback from the alternative.

We found that rhesus monkeys engaged in strategic exploration by decreasing their reliance on expected values (*random* exploration) when it was useful for the future (long horizon) and that active sampling was the only way to obtain information (partial feedback). Neurally, we found prefrontal strategic exploration signals in the anterior and mid-cingulate cortex (ACC/MCC) and dorsolateral prefrontal cortex (dlPFC). However, we did not find a significant modulation by the horizon and feedback type of the effect of uncertainty (*directed* exploration) on choices. When making choices in a sequence (long horizon), we found evidence that macaques used counterfactual feedback to guide their choices. Complementing this activity at the time of decision, in the complete feedback condition, we found overlapping chosen and unchosen outcome prediction error signals in the orbitofrontal cortex (OFC), at the time of receiving the outcome. The counterfactual prediction errors in the OFC are particularly interesting as they point to the neural system that allowed the macaques to forgo having to make exploratory choices in the complete condition, which could also change how the MCC-dlPFC network represented the value of the chosen option.

## Results

### Probing strategic exploration in monkeys

Three monkeys performed a sequential choice task inspired by Wilson and colleagues [12]. In this paradigm called the horizon task, monkeys were presented with one choice (short horizon) or a sequence of four choices (long horizon) between two options (Fig 1A). Each option belonged to one side of the screen and had a corresponding touch pad located under the screen (see Materials and methods for details). Both types of choice sequence (long and short horizon) started with an "observation phase" during which monkeys saw four pieces of information randomly drawn from both options and reflecting outcome distribution of each option. They received at least one piece of information per option (Fig 1B). Each piece of information was presented exactly like subsequent choice outcomes as a bar length (equivalent of 0 to 10 drops of juice) drawn from each option's outcome distribution. The animals had learned that the length of the orange bar on a yellow background indicated the number of drops of juice associated with that specific option on a given trial (Fig 1B). One option was associated with a larger reward (more drops of juice) on average than the other. The means of the distributions were fixed within a sequence but unknown to the monkey. Monkeys only received the reward associated with the option they chose at the end of each choice.

First, we manipulated whether the information gathered during the first choice could be useful in the future. During a session, we varied the number of times monkeys could choose between the options (horizon length). The horizon length was visually cued (Fig 1A and 1B). On *short horizon* trials, the information provided by the outcome of the choice could only be used for the current choice and was then worthless going forward. On *long horizon* trials, it could be used to guide a sequence of four choices. Second, we manipulated the contingency between choice and information by varying the type of feedback monkeys received after their active choices (the observation phase was identical for partial and complete feedback conditions). In the partial feedback condition, they only saw the outcome for the option they chose. In the complete feedback condition, they saw the outcome of both the option they chose and the alternative option (Fig 1D and 1E). In the latter case, the information about the options could be learned from the counterfactual outcomes—the outcome that would have been obtained had a different choice been made. This type of feedback is sometimes referred to as

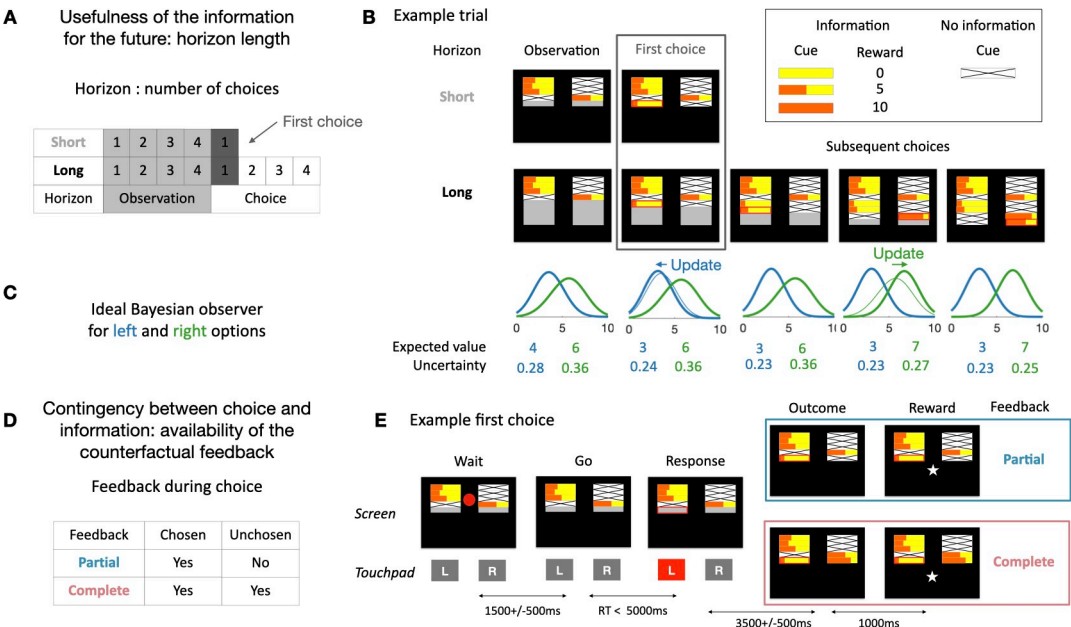

**Fig 1. Task and model.** (**A**) During the task, we manipulated whether the information could be used in the future by including both long and short horizon sequences. In both trial types, monkeys initially received four samples ("observations") from the unknown underlying reward distributions. In short horizon trials, they then made a one-off decision between the two options presented on screen ("choice"). In long horizon trials, they could make four consecutive choices between the two options (fixed reward distributions). On the first choice (highlighted), the information content was equivalent between short and long horizon trials (same number of observations), whereas the information context was different (learning and updating is only beneficial in the long horizon trials). (**B**) Example short and long horizon trials. The monkeys first received some information about the reward distributions associated with choosing the left and right option. The length of the orange bar indicates the number of drops of juice they could have received (0–10 drops). The horizon length of the trial is indicated by the size of the grey area below the four initial samples. The monkeys then make one (short horizon) or four (long horizon) subsequent choices. As monkeys progressed through the four choices, more information about the distributions was revealed. Displayed here is a partial information trial where only information about the chosen option is revealed. (**C**) Ideal model observer for the options of the example trial shown in B (color code corresponds to the side of the option). The distributions correspond to the probabilities to observe the next outcome for each option. The expected value corresponds to the peak of the distribution and the uncertainty to the variance. Thick lines correspond to post-outcome estimate and thin lines to pre-outcome estimates (from the previous trial). (**D**) We also modulated the contingency between choice and information by including different feedback conditions. In the partial feedback condition, monkeys only receive feedback for the chosen option. In contrast, in the complete feedback condition, they receive feedback about both options after active choices (not in the observation phase). (**E**) Example partial and complete feedback trials (both short horizon). Here, the observation phase shown in (B) is broken up into the components the monkeys see on screen during the experiment. Initially, the samples were displayed on screen, but a red circle in the center indicates that the monkeys could not yet respond. After a delay, the circle disappears, and the monkeys could choose an option. After they responded, the chosen side was highlighted (red outline). After another delay, the outcome was revealed. In the partial feedback condition (top), only the outcome for the chosen option was revealed. In contrast, in the complete feedback condition (bottom), both outcomes were revealed. After another delay, the reward for the chosen option was delivered in both conditions.

"hypothetical" [30] or "fictive" feedback [31]. The feedback condition was not cued but was fixed during a session.

To assess monkeys' sensitivity to the expected value and the uncertainty about the options, we set up an ideal observer Bayesian model (see Materials and methods for model details), which estimates the probability of observing the next outcome given the current information (Fig 1C). This model uses only the visual information available on the screen to infer the true underlying mean value of each options but does not use the horizon nor the feedback type as those were irrelevant for this inference. We extracted the expected value (peak of the probability distribution of the next observation, i.e., most likely next outcome) and the uncertainty (variance) of the options from the model to evaluate monkeys' sensitivity to these variables. If

monkeys did not engage in strategic exploration, the effect of expected value should be unaffected by the manipulations of horizon and feedback as was the case for the model.

## The horizon length and the type of feedback modulate monkeys' exploration

We first focused our analysis on the first choice of the trial, as the information about the reward probability of two options was identical across horizons and feedback conditions, such that choices should only be affected by the contextual manipulations (horizon and feedback type). If monkeys were sensitive to whether the information could be used in the future, they would explore more in the long compared to the short horizon. This is because information obtained early in a trial can only beneficial for subsequent choices in long horizon trials. Moreover, exploration should only occur when obtaining information is instrumentally dependent upon it, i.e., in the partial feedback condition (Fig 2A).

We first ensured that monkeys' choices were influenced by the expected value computed by the Bayesian model. We looked at the accuracy (defined as choosing the option with the highest expected value according to the model) during the first choice. For the two horizon lengths and in both feedback conditions, accuracy was above chance level ($t$ test compared to a distribution with a mean at 0.5; partial feedback short horizon: t (40) = 10, $p < 0.001$, partial feedback long horizon: t (40) = 8.930, $p < 0.001$, complete feedback short horizon: t (39) = 7.9, $p < 0.001$, complete feedback long horizon: t (39) = 8.963, $p < 0.001$, Fig 2B). Therefore, monkeys used the information provided by the informative observations on each trial to guide their choices. Monkeys also adjusted their choices to variations in expected value, as can be seen when pooling together both feedback conditions and horizon lengths (Fig 2C; see statistical significance in Fig 2D).

Although choices were guided by the expected value of the options above chance level, monkeys still sometimes chose the less valuable option in both conditions and horizons (Fig 2B and 2C). We examined whether monkeys were less driven by expected value on partial feedback long horizon trials, as exploration is only a relevant strategy on these trials (Fig 2A).

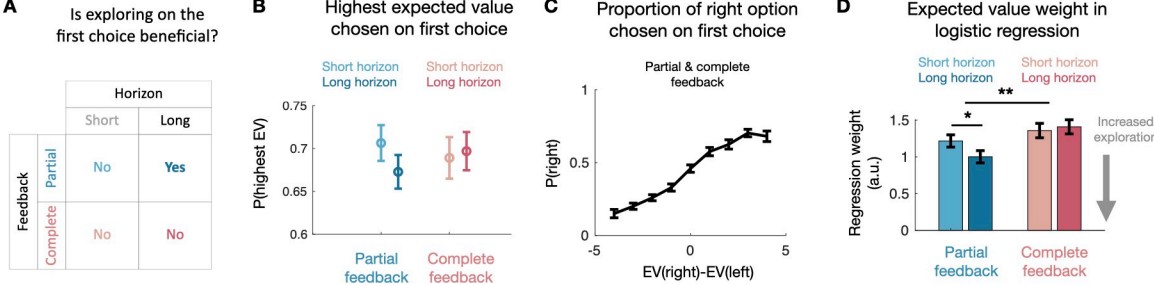

**Fig 2. First choice.** (**A**) In our experimental design, on the first choice of a horizon, *directed* exploration is only sensible in long horizon trials in the partial feedback condition. This is because in short horizon trials, the information gained by exploring is of no use for subsequent choices, so a rational decision-maker would only choose based on the expected value of the options. Moreover, in the complete feedback condition, all information is obtained regardless of which option is chosen, so an ideal observer would again always choose the option with the highest expected value. (**B**) The proportion of trials in which the monkeys chose the option with the higher expected value is above chance level (0.5) across both feedback conditions and horizons. Mean across sessions (partial feedback: 41 sessions, complete feedback: 40 sessions). (**C**) Monkeys' choices are sensitive to nuanced differences in expected value. Mean across all sessions (81 sessions). (**D**) According to the logistic regression model predicting monkeys' first choices in a horizon (see main text and methods for details), monkeys' first choices are less driven by expected value in the partial than in the complete feedback condition. Within the partial feedback condition, they are less driven by expected value in long than in short horizon trials. No such difference was found in the complete feedback condition. This is evidence that monkeys deliberately modulate their exploration behavior to explore more on partial feedback long horizon trials, where exploration is sensible (see (A)). Error bars indicate standard error to the mean in B and C and standard deviation in D. Data and code to reproduce the figure can be found at https://doi.org/10.5281/zenodo.7464572.

To test this hypothesis, we ran a single logistic regression predicting responses during first choices in the partial and complete conditions with the following regressors: the expected value according to our Bayesian model, the uncertainty according to our Bayesian model, the horizon (short/long), and the interactions of expected value and uncertainty with horizon. In the same model, we added two potential biases, a side bias and tendency to repeat the same action. We fitted regressors to vary by condition (partial or complete feedback) and by monkey, and modelled sessions as random effects for each monkey, with all regressors included as random slopes. We confirmed that in both feedback conditions, monkeys tended to choose the option with the highest expected value ($p < 0.000001$ in the partial condition and $p < 0.000001$ in the complete; one-sided test, based on sample drawn from Bayesian posterior; see Materials and methods). We identified that monkeys relied more on the difference in expected value in the complete than in the partial feedback condition ($p = 0.0024$; one-sided test), and in short horizon than in the long horizon in the partial condition only ($p = 0.0163$ in the partial condition and $p = 0.6598$ in the complete; one-sided test). Thus, animals engaged in strategic exploration by reducing their reliance on expected value. In other words, animals strategically modulated the degree to which they used *random* exploration both depending on the horizon length and feedback type (S3A Fig).

We next looked at the effect of uncertainty. Exploratory behaviors should be sensitive to how much they can reduce uncertain, i.e., the animals should optimally pick the most uncertain option when they explore [12]. We found that monkeys were sensitive to uncertainty overall, avoiding options that were more uncertain in the partial and the complete feedback conditions ($p = 0.0081$ in the partial condition and $p = 0.00025$ in the complete; one-sided test) (see S1 Fig for full model fit and the posteriors for each individual subject). This risk aversion was driven by the difference in number of information presented as when we restricted our analysis to the trials where they received 2 information about each option, monkeys showed a small preference for the more uncertain option ($p = 0.077$ in the partial condition and $p = 0.066$ in the complete; $p = 0.02$ when combined; one-sided test) (see S2 Fig for full model fit and the posteriors for each individual subject). However, we found no statistically reliable difference in the sensitivity to the uncertainty across the experimental conditions. We also ran a second model that used the number of available information (which is mathematically equivalent to the model used by Wilson and colleagues [12]) rather than uncertainty and found identical results, both for the effects on expected value and the absence of an effect on the sensitivity to the number of available information (S3B and S4 Figs for full model fit and the posteriors for each individual subject). Therefore, uncertainty did not play a key role in strategic exploration in our task. This indicates that our macaques did not use *directed* exploration to strategically guide their choices (S3 Fig).

Finally, we checked whether the decision variables were stable over trials and across sessions. First, in the above regression model, we added interaction terms with the trial number. We found no significant interaction with our regressors of interest (expected value, expected value interaction with horizon, uncertainty, and uncertainty interaction with horizon). Second, we fitted each session separately and looked for a linear trend in the session number. We found no linear nor clear trend for the regressors over sessions. Overall, we found no evidence that monkeys' decision variables changed throughout the recording.

## Monkeys learn from chosen and counterfactual feedbacks

We next assessed whether monkeys used the information they collected during their previous choices to update their choice, and how the nature of the feedback affected this process. To this end, we focused our analysis on choices from long horizon trials. On such trials, monkeys'

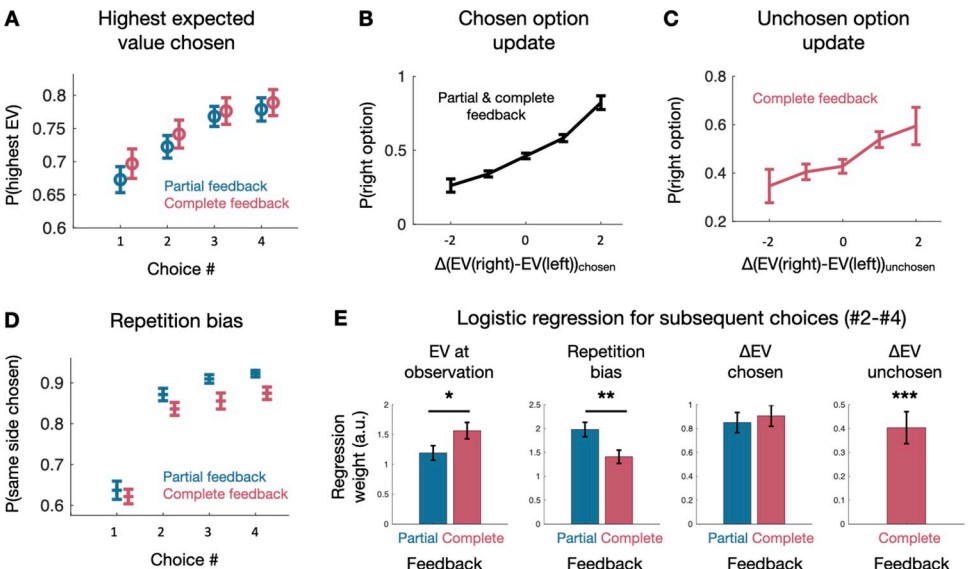

**Fig 3. Behavioral update.** (**A**) As monkeys progressed through the long horizon, they were more likely to choose the option with the higher expect reward in both the partial and complete feedback condition. Mean across sessions (partial feedback: 41 sessions, complete feedback: 40 sessions). (**B**) Monkeys were sensitive to changes in the expected value compared to the baseline expected value they experienced during the observation phase both for the chosen option (mean across all sessions (81 sessions)) and (**C**) the unchosen option (mean across all complete feedback sessions (40 sessions)). (**D**) Monkeys were also more likely to repeat their choice as they progressed through the long horizon. Mean across sessions (partial feedback: 41 sessions, complete feedback: 40 sessions). (**E**) Results of the single logistic regression model predicting second, third, and fourth choices in the long horizon. In both the partial and complete feedback, monkeys were sensitive to the expected value at observation but more so in the complete than the partial feedback condition (left). Monkeys tended to repeat previous choices in both conditions but more so in the partial than in the complete feedback condition (center left). In both conditions, monkeys were sensitive to the change in expected value compared to the observation phase with no significant difference between conditions (center right). In the complete feedback condition, monkeys were also sensitive to the change compared to baseline of the additional information they received. Error bars represent standard error to the mean in A–D and standard deviation in E. $^{*}p < 0.05$, $^{**}p < 0.01$, and $^{***}p < 0.001$. Data and code to reproduce the figure can be found at https://doi.org/10.5281/zenodo.7464572.

accuracy (defined as choosing the option with the highest expected value according to the model) was always above chance level ($t$ test compared against a mean of 0.5; all $p < 10^{-10}$) and increased as they progressed through the sequence ($t$ test compared against a mean of 0 of the distribution regression coefficients of the trial number onto the accuracy (both z-scored) for each session; partial feedback condition: t (40) = 11.3653, $p < 0.001$, complete feedback condition: t (39) = 5.6590, $p < 0.0001$) (Fig 3A). We inferred that this improvement was due to the use of the information collected during the choices. To examine this, we isolated the change in expected value compared to the initial "observation phase" (see Materials and methods). We found that monkeys were sensitive to the change in expected value both for the chosen option (in the partial and complete feedback conditions) and the unchosen option (counterfactual feedback in the complete feedback condition only) (Fig 3B and 3C; see statistical significance in Fig 3E). Monkeys displayed a significant tendency to choose the same option ($t$ test compared against a mean of 0.5; all $p < 10^{-6}$), which sharply increased after the first trial (paired $t$ test between the first choice and the subsequent choices; all $p < 10^{-10}$) and kept increasing after the first choice ($t$ test compared to a distribution with a mean at 0 of the distribution regression coefficients of the trial number onto the probability to choose the same option (both z-scored) for each session; partial feedback condition: t (40) = 5.3026, $p < 0.001$, complete feedback condition: t (39) = 3.1265, $p = 0.0033$) (Fig 3D).

We investigated the determinants of these effects by performing a single logistic regression for all non-first choices with the following regressors: the expected value and uncertainty during the observation phase (which served as a baseline for subsequent choices), the change in these baselines as new information was revealed as they progressed through the horizon. We also added in the same model three potential biases in choices: a side bias, the tendency to repeat the same action, and a bias for choosing the option most often chosen (see S5 Fig for full model fit and the posteriors for each individual subject). Just as with the previous regression model for first choices, we again allowed regressors to vary by condition and monkey and modelled sessions as random effects. We confirmed that monkey remained sensitive to the difference in expected value during the observation phase and that guided the first choice ($p < 0.000001$ in the partial condition and $p < 0.000001$ in the complete; one-sided test). Consistent with the choice behavior on the first choice, monkeys relied more on this difference in the complete than in the partial feedback condition in subsequent choices ($p = 0.0192$, one-sided test; Fig 3E). Monkeys were biased towards repeating the same choice ($p < 0.000001$ in the partial condition and $p < 0.000001$ in the complete; one-sided test), but this bias was also more pronounced in the partial feedback condition ($p = 0.0018$, one-sided test; Fig 3E) as can already be seen in Fig 3B. Monkeys also preferred to choose the option most chosen ($p < 0.000001$ in the partial condition and $p < 0.000001$ in the complete; one-sided test), which explained the increase in repetition bias over time, but this was not affected by the feedback type (partial > complete: $p = 0.309$) (S5 Fig). Monkeys were sensitive to the change in expected value when the information was related to the chosen option ($p < 0.000001$ in the partial condition and $p < 0.000001$ in the complete; one-sided test), with no statistical difference between the partial and complete feedback conditions (partial > complete: $p = 0.6913$). Finally, in the complete feedback condition, monkeys were sensitive to the change in expected values obtained from the counterfactual feedback ($p < 0.000001$; one-sided test; Fig 3E).

Overall, we found that on top of being more sensitive to the expected value difference during the initial evaluation, monkeys were less likely to be biased towards repeating the same action when they had counterfactual feedback to further guide their choices in the complete feedback condition. They were able to learn about the options, using both the chosen and the counterfactual feedback when it was available.

## Strategic exploration signals in ACC/MCC and dlPFC

To identify brain areas associated with strategic exploration, we ran a two-level multiple regression analysis using a general linear model (GLM). For each individual session, we used a fixed-effects model. To combine sessions and monkeys, we used random effects as implemented in the FMRIB's Local Analysis of Mixed Effects (FLAME) 1 + 2 procedure from the FMRIB Software Library (FSL). We focused our analysis on regions previously identified in fMRI studies on reward valuation and cognitive control in monkeys [24–29]. Thus, to only look at the regions we were interested in and to increase the statistical power of our analysis, we only analyzed data in a volume of interest (VOI) covering frontal cortex and striatum (previously used by Grohn and colleagues [29]). We used data from 75 (41 partial feedback and 34 complete feedback) of the 81 (41 partial feedback and 40 complete feedback) sessions we had acquired (fMRI data from 6 sessions were corrupted and unrecoverable). Details of all regressors included in the model can be found in the Materials and methods section. In addition to the analysis in the VOI, we examined the activity in the functionally and anatomically defined regions of interest (ROIs). These ROIs were not chosen a priori but were selected based on the activity in the VOI. The goal of these analyses was either (i) to examine the effect of a different variable than the one used to define the ROI in our VOI, which is an independent test so we

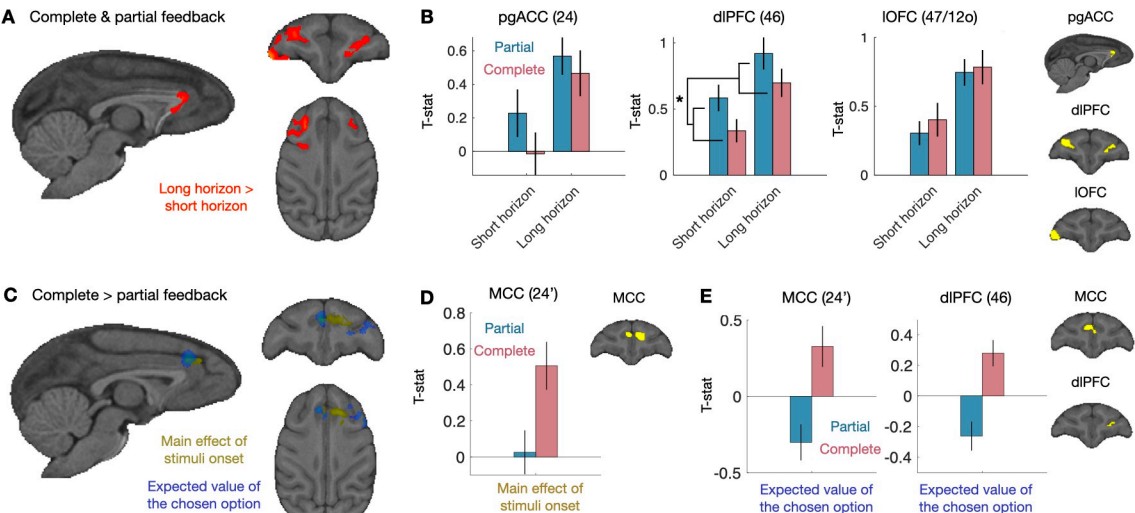

**Fig 4. First choice neural results.** (**A**) When combining partial and complete feedback sessions, we found clusters for a differential in activity in long horizon than short horizon in the pgACC, the dlPFC, and the lateral OFC. Cluster $p < 0.05$, cluster forming threshold of $z > 2.3$. (**B**) We placed ROIs (in yellow) in the overlap of the functional cluster and anatomical region and extracted t-statistics for the difference between long horizon and short horizon. Mean across sessions (partial feedback: 40 sessions, complete feedback 34 sessions). (**C**) We looked for differences in how the contingency between choice and information (complete vs. partial feedback) modulates the initial information that was presented before first choices. Within our VOI, we found clusters of activity in MCC both for the main effect of feedback type and a greater sensitivity to expected value in the complete feedback condition. We also found a cluster of activity in dlPFC for a greater sensitivity to expected value in the complete feedback condition. (**D**) We placed an ROI (in yellow) in the part of MCC that is activated by the main effect of feedback type and extracted the t-statistics of the regressor for every session. We found that the effect we observe in the VOI is driven by increased activity in the complete feedback condition, whereas there is no activity in the partial feedback condition. Mean across sessions (partial feedback: 40 sessions, complete feedback 34 sessions). (**E**) We also placed ROIs (in yellow) in the parts of MCC and dlPFC where we found significant clusters in the VOI for the interaction of feedback type and expected value and extracted the t-statistics for the expected value regressor of every session. Plotting these regressors separately for feedback type reveals that both MCC and dlPFC were more active when an option with high expected value was chosen in the complete feedback condition, whereas they were more active when an option with low expected value was chosen in the partial feedback condition. Mean across sessions (partial feedback: 40 sessions, complete feedback 34 sessions). Error bars represent standard error to the mean. *$p < 0.05$. Data and code to reproduce the figure can be found at https://doi.org/10.5281/zenodo.7464572. dlPFC, dorsolateral prefrontal cortex; MCC, anterior and mid-cingulate cortex; OFC, orbitofrontal cortex; pgACC, pregenual anterior cingulate cortex; ROI, region of interest; VOI, volume of interest.

could look for statistical significance of this different variable on the activity in the ROI, or (ii) to illustrate an effect revealed in the VOI, which is not an independent test, so we did not do any statistical analysis.

To examine how monkeys use initial information displayed during the observation phase of the task differently depending on the horizon and the feedback condition, we examined the brain activity when the stimuli were presented on the first choice ("wait" period; Fig 1D). Crucially, there was no difference in the visual inputs between the partial and the complete feedback condition, as the nature of the feedback was not cued and fixed for blocks of sessions and monkeys only received the counterfactual feedback after an active choice (not in the observation phase). We first investigated the main effects of our two manipulations: the overall effect of the horizon and feedback type on brain activity.

We combined all sessions and looked for evidence of different activations in the long and short horizon. We found a significantly greater activity for the long horizon in 3 clusters (cluster $p < 0.05$, cluster forming threshold of $z > 2.3$; Fig 4A, see S1 Table for coordinates of cluster peaks). One cluster was centered on the pregenual anterior cingulate cortex (pgACC), and the striatum and two clusters of activities were centered on the dlPFC and extended in the lateral orbitofrontal cortex (lOFC, area 47/12o; see Materials and methods for more details about

OFC subdivisions) with one on each hemisphere. In an independent test, we placed ROIs by calculating the functional and anatomical overlap for each Brodmann area 24, 46, and 47/12o and extracted the t-statistics of the regressor to examine the effect the contingency between choice and information (feedback condition). We observed no effect of the feedback type in ACC ($p = 0.19$) and lOFC ($p = 0.53$), but we found a main effect of feedback type in the dlPFC (two-way ANOVA, $F(144, 147) = 4.86$, $p = 0.029$) and no interaction anywhere (ACC: $p = 0.29$, dlPFC: $p = 0.9$ and lOFC: $p = 0.78$). This revealed that a subpart of the pgACC and the lOFC were sensitive to the horizon length, while the dlPFC showed an additive sensitivity to the horizon length and the feedback type, such that it was most activated in the long horizon and partial feedback, when exploration is beneficial.

We next examined the effect of the feedback in our VOI. We found one cluster around the MCC that was significantly modulated by the difference between the activity during the complete and partial feedback conditions during stimuli presentation on the first choice (Fig 4C, yellow contrast; see S1 Table for coordinates of cluster peaks). To examine this effect further, and although it is not an unbiased test, we defined an ROI by taking the overlap between our functionally defined cluster and Brodmann area 24′. Extracted the t-statistics of each session from the regressor from this ROI revealed that the MCC is more active at the time of choice in the complete feedback condition but not in the partial feedback condition (Fig 4D). We found no interaction between the horizon length and the feedback type in our VOI. Thus, a different subpart of the MCC that was sensitive to the horizon length was sensitive to the type of feedback.

Behaviorally, we observed that strategic exploration was implemented by decreasing the influence of expected value on the choice. We therefore next looked for evidence of stronger expected value signals in complete feedback condition compare to the partial feedback condition. We tested the expected value of the chosen option, the unchosen option, and the difference in expected values between the chosen and unchosen options. We only found activity related to the expected value of the chosen option. We found two clusters of activities bilaterally in the MCC (area 24′) and the left dlPFC (area 46) that were modulated by the contingency between choice and information (Fig 4C; see S1 Table for coordinates of cluster peaks). We again placed two ROIs by calculating the functional and anatomical overlap for Brodmann areas 24′ and 46 and extracted the t-statistics of the regressor. Although this is not an unbiased test, we can see that the MCC and dlPFC seemed to be active when an option with a low expected value was chosen, whereas in the complete feedback condition, they were more active when choosing high expected value options (Fig 4E for illustration). We found, however, no difference of the strength of this sensitivity between short and long horizons. Thus, we found that the availability of the counterfactual feedback in the complete feedback condition decreased—and potentially even inverted—the sensitivity of the MCC and dlPFC to the expected value of the chosen option. We conducted additional exploratory brain–behavior correlations but found no significant relationships to behavioral sensitivity (see "Author Response" file within the Peer Review History tab for additional details).

Finally, we looked for signals that were related to the expected outcome of the chosen option and that were common to both feedback conditions. Consistent with previous studies [32–35], when we combined the partial and complete feedback conditions session and took all trials in the "wait" period, we found a large activation related to the expected value of the chosen option (which is the same as the chosen action in our task) spanning from the motor cortex/somatosensory cortex, the dlPFC, the OFC, and striatum, as well as an inverted signal in the visual areas in the whole brain (without mask; S6A Fig). We also found a clear representation of the uncertainty about the chosen option on the first choice (when the magnitude of the uncertainty about the chosen option is equivalent in the partial and complete feedback

conditions as no counterfactual feedback has yet been provided) in the right medial prefrontal cortex (24c and 9m) that extended bilaterally in the frontal pole (10mr) (S6B Fig). We conducted additional exploratory brain–behavior correlations but found no significant relationships to behavioral sensitivity (see "Author Response" file within the Peer Review History tab for additional details).

Overall, we found that pgACC and MCC reflected the horizon length and the type of feedback, respectively. The dlPFC was linearly modulated both, with the strongest activation in the long horizon and partial feedback, when exploration is beneficial. Additionally, the feedback type modulated the effect of the chosen expected value on the activity of the MCC and the dlPFC, such that it was more active for low value choices only when obtaining information was contingent on choosing an option.

## Chosen and counterfactual outcome prediction error signals in the OFC

We next examined the brain activity when the outcome of the choice is revealed ("outcome" period in Fig 1D) and monkeys are updating their beliefs about the options. After the first choice, the sequences of events played out differently in the partial and complete feedback conditions; therefore, we analyzed each dataset separately in regard to feedback. At outcome, the partial feedback condition closely resembles previously reported results from fMRI studies in monkeys [27,29]. We looked for brain regions with an activity that was modulated by magnitude of the outcome prediction error signals, i.e., the difference between the outcome and the expectation (S7A Fig). Consistent with these studies, we found the expected clusters of activity in the medial prefrontal cortex and bilaterally in the motor cortex in our VOI (see fig for outcome only–related activity). When we time-locked our search to the onset of the reward (1 s after the display of the outcome, on a different GLM), we also found the classic prediction error related activity in the ventral striatum at the whole brain level (S7C Fig).

We then turned to the complete feedback condition, in which we simultaneously presented the outcome of the chosen and the unchosen after the first active choice, in order to examine the neural substrates involved in learning about counterfactual feedback and the extent to which they overlap with learning about chosen feedback. We looked in our VOI for brain regions with an activity that was modulated by the prediction error for the chosen option and the unchosen option. We found a cluster of activity around the lOFC (area 47/12o) that was negatively modulated by the prediction error for the chosen option and a cluster of activity around the medial orbitofrontal cortex (mOFC, area 14) that was negatively modulated by the prediction error of the unchosen option (Fig 5A; see S1 Table for coordinates of cluster peaks). These clusters intersected in the central part of the OFC (cOFC, area 13). Prediction error activity should show both an effect of outcome and expectation, with opposite signs. To independently test whether observed effects were prediction errors, rather than being driven by the outcome or the expectation alone, we extracted the t-statistics for both outcome and expectation in ROI defined by their outcome-related activity only and looked for a modulation by the expectations (S8 Fig). Again, we defined ROIs based the functional modulation by the magnitude of the chosen outcome and anatomical overlap. For the chosen outcome, we found that lOFC did not show a significant positive expectation for the chosen outcome ($p = 0.1083$) (Fig 5C). We found that the somatosensory cortex (area 3) showed a strong positive chosen outcome signal and as well as a positive modulation by the chosen expectation (T (33) = 2.5246, $p = 0.017$) and the ventrolateral prefrontal cortex (vlPFC) (area 45) had no sensitivity for the chosen expectation ($p = 0.95$) (S6B Fig). Using the same procedure with the unchosen outcome, we found that the cOFC showed a positive expectation about the unchosen outcome (t

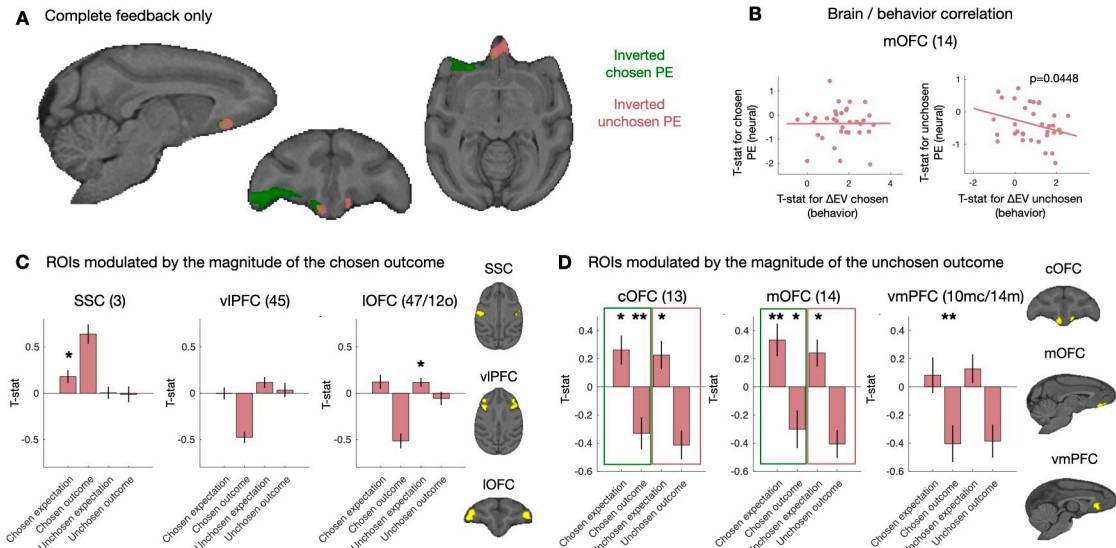

**Fig 5. Prediction error neural results.** (**A**) In complete feedback sessions only, we found clusters for inverted prediction error activity in the central part of OFC (area 13), extending into lOFC (area 47/12o). We also found inverted prediction error activity in the cOFC (area 13) and mOFC (area 14) for the unchosen, counterfactual reward. (**B**) Brain–behavior correlational analysis between the prediction error signal in the mOFC (t-statistic) and session-specific t-statistic of the behavioral effect of the change in expected value on choices (estimated with a separate GLM for each session). (**C**) We placed ROIs (in yellow) in the overlap of the functional cluster modulated by the magnitude of the chosen outcome and anatomical region. We extracted t-statistics for reward and expectation, both of the chosen and unchosen option. Prediction error activity should evoke both a reward and an expectation response with opposite signs. We did not found evidence for outcome expectation of the chosen option. Mean across complete feedback sessions (34 sessions). (**D**) When defining the ROIs (in yellow) according to the response to the magnitude of unchosen outcome, we find evidence for a classic reward prediction error and a counterfactual prediction error about the unchosen option both in cOFC and mOFC: We observe activity related both to the obtained and the unobtained reward, and also activity related both the chosen and unchosen outcome expectation. Mean across complete feedback sessions (34 sessions). Error bars represent standard error to the mean. $^*p < 0.05$, $^{**}p < 0.01$. Data and code to reproduce the figure can be found at https://doi.org/10.5281/zenodo.7464572. cOFC, central OFC; GLM, general linear model; OFC, orbitofrontal cortex; lOFC, lateral OFC; mOFC, medial OFC; ROI, region of interest.

(33) = 2.2617, $p$ = 0.0304), as well as a negative modulation by the chosen outcome (t (33) = −2.8761, $p$ = 0.007) and a positive modulation by the expectation about the chosen outcome (t (33) = 2.5560, $p$ = 0.0154). We found a similar pattern in the mOFC (unchosen expectation: t (33) = 2.5130, $p$ = 0.017; chosen outcome: t (33) = −2.2455, $p$ = 0.0316; chosen expectation: t (33) = 2.8729, $p$ = 0.0071). The ventral-medial prefrontal cortex (area 10m according to the atlas we used [36] but has been called 14m [37]) showed a negative modulation of its activity by the unchosen and the chosen (t (33) = −3.079, $p$ = 0.004) outcomes but no sensitivity to the expectations. Overall, we found that the cOFC and mOFC both showed prediction error–related activity for both the chosen and the unchosen outcomes, and with the same sign.

To test the OFC prediction error effects even further, we ran an exploratory correlational analysis, between the ROIs based the prediction error signal (t-statistic) and session specific t-statistic of the behavioral effect of the change in expected value on choices (estimated with a separate GLM for each session with the same regressors as in Fig 5B). We wanted to see whether the strength of the counterfactual outcome prediction error in the brain is predictive of how much an animal uses it in a particular session. Only in mOFC (and not cOFC) did we see the expected—albeit modest—correlation between increased negative counterfactual prediction error signals and increased behavioral impact of the counterfactual information (Fig 5D, ß = −0.1701 ± 0.0971, t (32) = −1.7507, $p$ = 0.0448, one-sided test).

## Discussion

Weighing up exploration to gather new information with exploitation of your current knowledge is a key consideration for humans and animals alike. Inspired by recent work carefully dissociating value-driven exploration from simple lack of exploitation [12], we designed the horizon task to look at the behaviors and neural correlates of goal-directed evaluation of strategic exploration in rhesus monkeys. While strategic value-driven exploration is important to optimize the behavior in time, it is equally important to be able to learn from observations related to choices not taken. In particular, being able to process counterfactual information during learning is key to optimize exploration for only the kind of situations when active sampling is necessary.

### Strategic exploration as a reduction of the effect of expected value on choices

We know that monkeys can seek information before committing to a choice or to increase confidence about their decision [38–40]. However, here, we showed that monkeys could identify situations in which a strategic exploratory choice would lead to gaining information that would be beneficial for future decisions. Indeed, their choices were least influenced by expected value in the long horizon partial feedback condition, which is when there should be a drive to explore. This suggests that monkeys had a representation of the significance of the information and used it to plan future actions. Our results demonstrate that they could discern both whether information will be useful in the future (greater exploration in long horizon) and that choosing an option is instrumental to get information about it (greater exploration in the partial feedback condition).

Exploration during value-based decision-making has been conceived in different ways in the past. A simple way to account for exploration is the "epsilon-greedy strategy," in which a small fraction of choices is made towards the non-most rewarded option [1,3]. Along the same line, another way to formalize exploratory choices is through the noise or (inverse) temperature in the *softmax* choice-rule, which predicts that there are more exploratory choices when the expected values of the options are close [1,3–6]. This process is also called *random exploration* because the relaxation of the effect of expected value on choices could allow for stumbling upon better options by chance [12,15]. This form of exploration is negatively correlated with accuracy. Therefore, without varying the other features such as the usefulness of information for the future and the contingency between choice and information, it is impossible to know whether monkeys made a mistake or were exploring the non-most rewarded option to obtain information about it.

Here, we show that monkeys, like humans, can perform sophisticated choices that take into account the prospective value of discovering new information about the options [41]. Our results revealed that foraging behaviors in macaques do not only rely on simple heuristics (e.g., win-stay/lose shift) but are also based on strategic exploration. To some extent, it mirrors anticipatory switch to exploitative behavior once enough information has been learned about the information, even when the expected outcome has not yet been obtained [42]. It also adds to recent works showing that complex sociocognitive processes thought to be uniquely human such as mentalizing or recursive reasoning could be identified in rhesus monkeys [43,44]. However, contrary to human behavior, our monkeys only adapted by reducing their reliance on expected value (i.e., exploitative value) on choices, which corresponds to the degree to which they use *random* exploration. Humans also increase their preference for the most uncertain option when exploration is useful for the future, which has been referred to as *directed* exploration [12]. Our results can also be compared to recent work in monkeys showing that

monkeys choose novel options until they have an estimate of its value compare to other options [7–9], which can be interpreted as exploration for uncertainty reduction. Here, we found that monkeys did not seek to reduce overall uncertainty. This perhaps is because the number of reward samples was negatively correlated with uncertainty, and thus no option was novel in the sense that there was no information about it. Uncertainty-related behavior was not modulated by the feedback type nor the horizon, hinting that uncertainty (rather than novelty) was not a driver of strategic exploration. This suggests species specificity in exploratory strategy. In the future, a variation of our task could be used to test the effect of novelty on strategic exploration by offering zero information about one option.

## Use of counterfactual feedback in subsequent choices

We next investigated whether and how the availability of the counterfactual feedback impacted their subsequent choices in the long horizon. First, we found that having more information about the options in the complete feedback condition improved accuracy. In general, monkeys were more sensitive to the initial expected values of the options when there was no contingency between choice and information, in the complete feedback condition, but then utilized the feedback about the chosen option to the same degree in both conditions. However, in the complete feedback condition, monkeys additionally used the counterfactual feedback about the unchosen option to update their preference. Our results confirm that rhesus macaques are sensitive not only to direct reinforcers (i.e., the reward they obtain) but also counterfactual information [28,30,31,45]. Here, we clearly demonstrate that monkeys can learn directly from counterfactual feedback, via prediction error. We also demonstrated this learning is associated with OFC activity. Identifying that an alternative action could have led to a better outcome and acting upon it has been shown to modulate OFC activity in rodents [46], suggesting that this ability was present in the last common ancestor to primates and rodents 100 M years ago.

The availability of counterfactual feedback also helped compensate for the repetition bias that monkeys displayed during the performance of the sequence. This form of engagement can also be considered in terms of the exploration/exploitation trade-off, where exploiting corresponds to staying with the current or default option: Monkeys committed to an option at the beginning of the trial and only changed option if there were sufficient evidence that it was worth it. Consistently, humans and animals show a tendency to overexploit compared to the optimal policy in various tasks [47–49]. In general, there seems to be a cost associated with switching from the ongoing representation or strategy to a new one [50,51]. In our task, switching options also requires minimal physical effort as the monkeys are positioned in the sphinx position in the scanner. Additional information about the options, particularly about the alternative option, seems to encourage the reevaluation of the default strategy of persevering with the current option, enhance behavioral flexibility, and increases the willingness to bear the cost associated with the physical resetting required by switching target.

## Strategic exploration signals in ACC/MCC and dlPFC

Using fMRI, we investigated the neural correlates of the assessment of the possibility to use the information collected during the choice in the future, manipulated through horizon length, as well as the assessment of the contingency between choice and information, manipulated through the availability of the counterfactual outcome. Modulation of activity associated with exploratory behavior in an uncertain environment has been recorded in humans and monkeys in both the ACC and the MCC [42,52–54], but here, we found interesting anatomical distinctions. We found that the pgACC was more active when the information could be used in the

future in the long horizon. In humans, the pgACC activity has been shown to scale with the use prospective value to guide choices [41]. Thus, the pgACC might be critical to organize the behavior in the long run, beyond the immediate choice. The activity of a separate anatomical region of the MCC was modulated by the feedback type. The MCC has been shown to encode the decision to obtain information about the state of the world [55] and to integrate information about the feedback to adapt the behavior [56]. Here, we show that activity in the MCC was modulated prospectively by the feedback type. This activation was greater when more information was going to be provided, i.e., in the complete feedback condition. Thus, the MCC could be involved in anticipating more learning or regulating exploration prospectively based on the feedback that will be received. Critically, the dlPFC displayed an additive effect of the usefulness of exploration for the future and the contingency between choice and information. It was most active when both were true, and exploration was sensible. Moreover, in the complete feedback condition, the MCC and the dlPFC were more active when the expected value of the chosen option was high. Such modulation is in line with studies in monkeys and humans showing that neuronal activity in the MCC and the dlPFC does correlate with actions' values [57–60]. When the unchosen outcome was not available, MCC and dlPFC were more active when the expected value of the chosen option was low, which is consistent with the pursuit of an exploration strategy. Overall, the coordinated roles of ACC and MCC participate to the regulation of exploratory/exploitative behaviors, not only in rhesus macaque but also in humans [61].

Computational modelling of the activity in ACC/MCC and dlPFC suggest that ACC/MCC could regulate decision variables in the dlPFC based on the strategic assessment for exploration [62]. Noradrenaline has been shown to modulate the noise in the decision process that could fuel *random* exploration and potentially give a mechanism for modulation of exploratory activity [13,49,63,64]. Importantly, ACC/MCC also more generally interacts with other frontal lobe regions as well as monoamine systems and, in particular, the noradrenergic system making it a feasible mechanism for changing exploratory behaviors [65]. Specifically, a network consisting of the MCC, the dlPFC, and, potentially, the locus coeruleus could support the relaxation of the effect of expected value on choices based on the context. Altogether, these results illustrate how ACC/MCC and dlPFC might dynamically switch modes to pursue different goals depending on the task demands [42,52,54]. Future studies will aim at testing whether switching mode is dependent on noradrenergic inputs and which causal role both regions play in changing into and out of strategic exploration.

## Update signals for chosen and counterfactual outcomes in OFC

Being able to process counterfactual information during learning is key to reducing costly exploration to only the kind of situations when active sampling is necessary. Doing so requires an ability to process abstract information and learn from it similarly to experienced outcomes, without confusion between the two, which our monkeys achieved. Neurally, we found classic activations in the partial feedback condition in response to the magnitude of the outcome and the prediction error of the chosen option. At reward delivery, we observed a prediction error signal in ventral striatum, which has previously been reported in neurophysiological studies [66]. We also observed prediction error activity at outcome in MCC, which had been shown previously in neurophysiological recordings [67]. We also found that the prediction error for the chosen outcome modulated the activity of the OFC, but further examination showed that the lOFC was mostly sensitive to the chosen outcome. Previous studies have shown the lOFC involvement in learning and using choice–outcome associations to guide behavior [68–70], and causal studies demonstrated its role in credit assignment [71–74]. Here, in the presence of two outcomes—two stimuli—OFC could be crucial to integrate the information specifically

related the chosen option. We also revealed that the cOFC and mOFC carried clear chosen prediction error signals.

However, our results go beyond chosen prediction error signals and add two additional dimensions to our understanding the neural processing of counterfactual information during exploration and learning. Firstly, we were able to map out for the first time counterfactual prediction error signals in monkeys in the cOFC and mOFC. Importantly, by using fMRI, we could establish its specificity within the prefrontal cortex. In particular, we found signals for that counterfactual and the chosen outcomes but not the expectations in the vmPFC (10/mc14m). This adds to our knowledge of modulation of activity by counterfactual outcomes in gambling tasks had been reported in macaque lateral prefrontal cortex, MCC, and OFC [30,31]. Secondly, we found in the mOFC a relationship between the strength of the counterfactual prediction error signal and the extent to which the counterfactual outcomes influenced future choices (Fig 5). Encoding of the counterfactual outcome has also been observed in humans mOFC [23,75,76], and lesion of the mOFC in patients had been associated with an inability to use counterfactual information to guide future decisions [77]. Those results are compatible with the proposed broader role of the mOFC in representing abstract values [78,79]. Here, we show that it represents the comparison of the obtained counterfactual information with the expected counterfactual information. We found that the representation of the prediction error for the chosen and unchosen outcomes had the same sign at the time of outcome, which leads us to postulate that this update mechanism is independent of the frame of the decision [23,79,80].

Having identified the orbitofrontal source of counterfactual prediction errors in macaques opens up further possibilities to directly interfere with the neural processes in each system to see the effect it has on this complex adaptation of the animals' exploratory strategy. Furthermore, knowing how the brains of non-human primates might solve this complex sequential exploration task also sheds light on the building behavioral and neural blocks of reward exploration, learning, and credit assignment.

## Conclusions

Here, we showed that monkeys are able to assess the contingency between choice and information and the utility of information for the future when making strategic exploratory decisions. Different subparts of the ACC and MCC related to the assessment of these variables for strategic exploration, and the dlPFC represented them both additively, such that it was most active when exploration was beneficial. Only when the only way to obtain information was to explore did MCC and dlPFC show increased activity with less exploitative choices. This suggests a role in suppressing expected value signals when value-guided exploration should to be considered. Importantly, to limit costly exploration to when it is necessary being able to process counterfactual information is key. We showed monkeys could do this potentially by representing chosen and unchosen reward prediction errors in central and medial OFC. Furthermore, the strength of this signal in the mOFC was shown to be correlated with future decisions taken. Overall, our study shows how ACC/MCC-dlPFC and OFC circuits together might support exploitation of available information to the fullest and drive behavior towards finding more information when it is beneficial.

## Materials and methods

### Ethics statement

Animals were kept on a 12-h light–dark cycle, with access to water for 12 to 16 h on testing days and with ad libitum water otherwise. Feeding, social housing, and environmental enrichment followed guidelines of the Biomedical Sciences Services of the University of Oxford. All procedures were conducted under licenses from the United Kingdom (UK) Home Office in

accordance with The Animals (Scientific Procedures) Act 1986 and the European Union guidelines (EU Directive 2010/63/EU).

## Monkeys and task

Three male rhesus monkeys were involved in the experiment (Monkey M: 14 kg, 7 years old, monkey S: 12 kg, 7 years old, and monkey E: 11 kg, 7 years old). During the task, monkeys sat in the sphinx position in a primate chair (Rogue Research, Petaluma, CA) in a 3T clinical horizontal bore MRI scanner. They faced an MRI-compatible screen (MRC, Cambridge) placed 30 cm in front of the animal. Visual stimuli were projected on the screen by an LX400 projector (Christie Digital Systems). Monkeys were surgically implanted under anesthesia with an MRI-compatible cranial implant (Rogue Research) in order to prevent head movements during data acquisition. Two custom-built infrared sensors were placed in front of their left and right hands that corresponded to the stimuli on the screen. Blackcurrant juice rewards were delivered from a tube positioned between the monkey's lips. The behavioral paradigm was controlled using Presentation software (Neurobehavioral Systems, CA, USA).

The task consisted of making choices between two options by responding on either the left or right touch sensor to select the left or right stimulus, respectively. A trial consisted of a given number of choices (determined by the horizon length) between these two options (Fig 1A). Each option corresponded to one side for the entire trial (Fig 1B). After each choice, monkeys received a reward associated with the chosen option (Fig 1E). The reward was between 0 and 10 drops (0.5 mL of juice per drop) and was sampled from a Gaussian distribution with a standard deviation of 1.5 and mean between 3 and 7. The means of the underlying distribution were different for the two options and remained the same during a trial, such that one option was always better than the other. After each choice, monkeys also received a visual feedback on the reward (Fig 1E). This feedback was in the form of an orange rectangle displayed in a yellow rectangular window, such that the wider the orange rectangle, the greater the amount of juice (Fig 1B). It remained on the screen for the remainder of the trial.

At the beginning of each trial, prior to making their first choice, monkeys received 4 informative observations in total, which consisted of information about the reward they would have received if they had chosen the option (Fig 1B). This was displayed in the same manner as reward feedback and also remained on screen during the duration of the trial. For each informative observation, a non-informative observation was presented for the other option (Fig 1B). The non-informative observation was a white rectangle crossed by black diagonals. Half of the trials started with an equal amount of information about the two options (2 informative and 2 non-informative observations for each option), and the other half with an unequal (3 informative and 1 non-informative observations). The order and side were randomly determined.

A critical parameter was the number of choices in each trial (horizon length). In **short horizon** trials, monkeys were only allowed 1 choice before a new trial with new stimuli started, whereas in **long horizon** trials, they were allowed to make 4 choices between the options. Horizon conditions were blocked (5 consecutive trials of equal horizons) and alternated in the session. A second key manipulation was whether feedback was received only for the option they chose (**partial feedback** condition) or whether they received information about both the reward they received for the chosen option *and* the reward they would have received for selecting the alternative option (**complete feedback** condition) (Fig 1E).

A trial would proceed as follows (Fig 1E; timings in Table 1): After an inter-trial interval during which the screen was black, the stimuli were displayed, consisting of a large grey rectangle and the 4 horizontal bars of feedback information (Fig 1B and 1E). The length of the grey rectangle corresponded to the length of the horizon, which each line corresponding to a

**Table 1. Timings.**

| Choice | ITI | Go delay | Outcome delay | Reward delay | ICI |
|---|---|---|---|---|---|
| First | 4,000 ± 1,000 ms | 1,500 ± 500 ms (1,500 ± 200 ms for monkey E) | 3,500 ± 500 ms | 1,000 ms | 2,500 ± 500 ms |
| Second to fourth | N/A | | 1,500 ± 500 ms | | 1,500 ± 500 ms |

ICI, Inter-choice interval; ITI, Inter-trial interval.

choice, simulated or actual. Informative or non-informative stimuli were displayed on the first four lines. After the display of the stimuli, a red dot at the center of the screen disappeared, and monkeys were then allowed to choose between the two options by touching the corresponding sensor (in less than 5,000 ms or the trial restarted). A red rectangular frame appeared around the line on the side of the chosen option. After a delay, the outcome—the reward feedback—was displayed inside the rectangle. In complete feedback condition only, the reward that would have been gained on the other side (informative stimulus) was also displayed at the same moment. After an additional delay, a white star appeared on the screen, and the reward was delivered. After the end of the reward delivery, the star disappeared. In short horizon blocks, a new trial started after the inter-trial interval delay. In long horizon trials, the red dot appeared and then monkeys could choose among the options. The events leading to the reward were similar than for the first choice, but the delays were shorter. At the end of the fourth choice, a new trial started. The feedback condition monkeys were in was not explicitly cued but instead fixed both within and across several sessions (6 to 10 consecutive sessions). Sessions after a switch from one feedback condition to the other were included in the analysis since it only took one choice for monkeys to know the feedback condition.

Monkey M performed 14 sessions in the partial feedback condition and 13 (2 were corrupted and unrecoverable for fMRI analysis) in the complete feedback condition; monkey S performed 13 and 12 (3 corrupted sessions) sessions in each condition, respectively; and monkey E performed 14 and 15 (1 corrupted session) sessions in each condition. Sessions with less than 50 trials completed for the horizon task or with more than 80% bias for one side were removed from the analyses.

## Training

All monkeys followed the following training procedure, which lasted several months in a testing room mimicking the actual scanner room: First, they learned the meaning of the informative observation stimuli by choosing between a rewarded (1 to 10 drops) and a non-rewarded (0 drop) observation stimulus and then between different rewarded (0 to 10 drops) observation stimuli. They next learnt to associate an option with an expected value by choosing between a non-rewarded option (0 drop) and a rewarded option and then between rewarded options in the long horizon and partial feedback condition. We then introduced blocks of small and long horizon trials. Monkeys were then tested in the scanner room. They all had previous experience of awake behaving testing in the scanner. We discarded the first scanning session and then analyzed the following ones if they corresponded to our inclusion criterions in terms of number of trials and spatial bias. Monkeys M and S were introduced to the complete feedback condition during the training procedure; monkey E experienced it for the first time during testing.

## Bayesian expectation model

We analyzed the behavior using an ideal Bayesian model, which estimated the most likely next outcome given the previous observations about the options. Outcomes were randomly drawn

form a distribution of mean $\mu$ and fixed standard deviation. $P(x|\mu)$ is the probability that an outcome x would be observed given that it came from a distribution of mean $\mu$. Since outcomes were independently dram from a distribution of mean $\mu$, the probability of observing a set of outcomes $\{x_1 \ldots x_n\}$ was:

$$P(\{x_1 \ldots x_n\}|\mu) = \prod_{i=1}^{n} P(x_i|\mu) \tag{1}$$

Using Bayes' rule, we computed the probability that this observation was generated by a distribution of mean $\mu$:

$$P(\mu|\{x_1 \ldots x_n\}) = \frac{P(\{x_1 \ldots x_n\}|\mu)P(\mu)}{P(\{x_1 \ldots x_n\})} = \frac{\prod_{i=1}^{n} P(x_i|\mu)}{\sum_{j}^{N} \prod_{i}^{n} P(x_i|\mu_j)} \tag{2}$$

For each observation, we computed the probability of a new observation:

$$P(x_{n+1}|\{x_1 \ldots x_n\}) = \sum_{j=1}^{N} P(x_{n+1}|\mu_j)P(\mu_j|\{x_1 \ldots x_n\}) \tag{3}$$

Thus, we can compute the probability distribution of the future outcomes given a set of observations (Fig 1C shows how the distributions change with new observations).

In our model, the expected value (EV) of an option is the mean of the probability distribution of the set of observations. The uncertainty (U) about what the next outcome was represented by the variance of the distribution. In general, the more informative observations the subject has access to for an option, the closer the expected value to the actual mean of the underlying distribution and the smaller the uncertainty about this quantity. The weight of the expected value controls a specific form of exploration: the *reward-based* exploration. Reducing this parameter allows exploring options by relaxing the tendency to choose the most rewarded option.

## Choice model fit

We first focused our analysis on the first choice of the trial because it was similar in terms of information content (4 informative observations) across horizon lengths and feedback conditions. Contrary to subsequent choices in the long horizon, the expected value and the uncertainty about the expected value (which decreases with the number of informative observations, from 1 to 3) associated with each option were uncorrelated on the first choice. Indeed, in the partial feedback condition, if the option with the higher expected value is chosen more often, the uncertainty about its expected value decreases specifically, inducing a correlation between expected value and uncertainty about it. For these first trials $t$, we model the probability of picking the option that is presented on the right side of the screen as

$$P(right_t) = \sigma(b_{SB} + b_{RB}RB_t + b_{horizon}horizon_t + b_{EV}EV_t + b_{U}U_t + b_{ERhorizon}ER_t horizon_t + b_{Uhorizon}U_t horizon_t) \tag{4}$$

using logistic regression. Here, $\sigma$ is the sigmoid function, $RB_t$ is a categorical predictor that control for a repetition bias, $EV_t$ and $U_t$ denote the difference between the expected value / uncertainty of the options on the right and left side of the screen, $horizon_t$ is a categorical predictor for whether trial $t$ is a short or long horizon trial, and $e_t$ is the residual.

For the remaining trials (second, third, and fourth choice in the long horizon), we are interested in whether the animals change their behavior as new information becomes available. We

model these trials as

$$P(right_t) = \sigma(b_{SB} + b_{RB}RB_t + b_{\Delta Chosen}\Delta Chosen_t + b_{baselineER}baselineEV_t + b_{\Delta ERchosen}\Delta EVchosen_t$$
$$+ b_{baselineU}baselineU_t + b_{\Delta Uchosen}\Delta Uchosen_t + b_{\Delta ERunchosen}\Delta EVunchosen_t$$
$$+ b_{\Delta Uunchosen}\Delta Uunchosen_t)(5)$$

For this logistic regression, we used an additional bias, $\Delta Chosen$, which corresponds to the number of times the option on the right was chosen during the trial. Here, $baselineEV_t$ and $baselineU_t$ are the difference between the expected value/uncertainty of the right and the left option at the first trial within the horizon. As such, these regressors capture the impact the initial information displayed on screen has on subsequent choices. $\Delta EVchosen_t$ and $\Delta Uchosen_t$ capture the difference between the initial baseline and the information presented at the current trial based on the choices the animal has made; i.e., these regressors capture the update of outcome expectation and uncertainty between the right and the left option compared to the first choice based on the consequent rewards the animals experienced.

In our complete feedback condition, the animals can also learn about the reward they would have gotten, had they chosen the other option. This is not captured by $\Delta EVchosen_t$ and $\Delta Uchosen_t$ as these regressors only take the experienced (i.e., obtained) reward into account. To see how the unobtained reward affects choices, we included the regressors $\Delta EVunchosen_t$ and $\Delta Uunchosen_t$. These regressors are computed as the difference between the full outcome expectation and uncertainty (based on both the obtained and unobtained reward), and the outcome expectation and uncertainty for the obtained reward only. Just as with the $\Delta EVchosen_t$ and $\Delta Uchosen_t$, these regressors are also again constructed as the difference between the right and left option, and with the baseline subtracted.

To fit these models, we used STAN (https://mc-stan.org) and brms with the default priors [81,82]. For each model, we ran 12 chains, each with 1,000 iterations after a warm-up of 1,000 samples. We allowed all regressors to vary by condition (partial and complete) and animal (3 animals) as fixed effects. We modelled testing sessions as random effects with different Gaussians for each animal; i.e., for each regressor and each animal, we estimated the Gaussian distribution that session-level regressors are most likely drawn from. Group-level estimates of the coefficients were obtained by averaging across animals and/or conditions. To determine statistical significance, we counted the number of samples of the posterior that are greater/smaller than 0.

## MRI data acquisition and pre-processing

Imaging data were acquired using a 3T clinical MRI scanner and an 8-cm-diameter four-channel phased-array receiver coil in conjunction with a radial transmission coil (Windmiller Kolster Scientific, Fresno, CA). Structural images were collected under general anesthesia, using a T1-weighted MP-RAGE sequence (resolution = 0.5 × 0.5 × 0.5 mm, repetition time (TR) = 2.05 s, echo time (TE) = 4.04 ms, inversion time (TI) = 1.1 s, flip angle = 8˚). Three structural images per subject were averaged. Intramuscular injection of 10 mg/kg ketamine, 0.125 to 0.25 mg/kg xylazine, and 0.1 mg/kg midazolam were used to induce anesthesia. fMRI data were collected while the subjects performed the task with a gradient-echo T2* echo planar imaging (EPI) sequence (resolution = 1.5 × 1.5 × 1.5 mm, interleaved slice acquisition, TR = 2.28 s, TE = 30 ms, flip angle = 90˚). To help image reconstruction, a proton density–weighted image was acquired using a gradient-refocused echo (GRE) sequence (resolution = 1.5 × 1.5 × 1.5 mm, TR = 10 ms, TE = 2.52 ms, flip angle = 25˚) at the end of the session.

fMRI data were pre-processed according to a dedicated non-human primate fMRI pre-processing pipeline [29,83,84] combining FSL, Advanced Normalization Tools (ANTs), and

Magnetic Resonance Comparative Anatomy Toolbox (MrCat; https://github.com/neuroecology/MrCat) tools. In brief, T2* EPI data were reconstructed using an offline SENSE algorithm (Offline_SENSE GUI, Windmiller Kolster Scientific, Fresno, CA). Time-varying spatial distortions due to body movement were corrected by non-linear registration (restricted to the phase encoding direction) of each slice and each volume of the time series to a reference low noise EPI image for each session. The distortion corrected and aligned session-wise images were first registered to the animal structural image and then to a group-specific template in CARET macaque F99 space. Finally, the functional images were temporally filtered (high-pass temporal filtering, 3-dB cutoff of 100 s) and spatially smoothed (Gaussian spatial smoothing, full-width half maximum of 3m).

## fMRI analysis

We conducted our fMRI analysis using a hierarchical GLM (FSLREF). Specifically, we first fitted each individual session (in session space) using FSL's fMRI Expert Analysis Toolbox (FEAT). We then warped the session-level whole-brain maps into F99 standard space, before combining them using FEAT's FLAME 1 + 2 random effects procedure. Here, we used contrast to obtain separate estimates for the partial and complete sessions, the difference between partial and complete sessions and their average. To determine statistical significance, we used a cluster-based approach with standard thresholding criteria of $z > 2.3$ and $p < 0.05$. To increase power, we ran this cluster correction only in an a priori mask of the frontal cortex that was previously used in Grohn and colleagues [29].

On the session level, we included 58 regressors for the partial feedback sessions and 73 (including the same 58 regressors as in the partial feedback condition) for the complete feedback sessions. On top of these regressors, we also included nuisance regressors that indexed head motion and volumes with excessive noise. All regressors were convolved with an HRF that was modelled as a gamma function (mean lag = 3, standard deviation = 1.5 s), convolved with a boxcar function of 1 s.

The two main periods of the task we were interested in were when the stimuli first appeared on screen and when the outcome appeared on subsequent choices in the long horizon trials. At stimulus onset, we included a constant and regressors for the expected value of the chosen and unchosen options and also regressors for the uncertainty of the chosen and unchosen option. To allow us to examine the effects of these five regressors on first choices in short and long horizons and subsequent choices within the long horizon, we up each regressor by horizon and choice number (first choice short horizon, first choice long horizon, second choice long horizon, third choice long horizon, and fourth choice long horizon) for a total of 25 regressors. At outcome, we included another constant, the expected value of the chosen and unchosen options, the reward obtained on this trial, the absolute value of the prediction error of this trial (|reward—expected value|), and the update in uncertainty on this trial. Again, all of these regressors were split up by horizon and choice number, for a total of 30 regressors at outcome. On top of these regressors of interest, we also included 3 control regressors: the log response time at stimulus onset, a constant at decision, and the response side (left or right) at decision. In the complete feedback condition, we included additional regressors: At outcome, we added regressors for the reward of the unchosen option, the absolute prediction error for the unchosen option, and the update in uncertainty for the unchosen option. Splitting these regressors up by horizon and choice within a horizon yields an additional 15 regressors.

Having split up all regressors this way into choice horizon and number of choices within a horizon, we used planned contrasts combining them again to answer our questions of interest. At stimulus onset, we were only interested in first choices, as this allowed us to compare

whether the animals represented expected value and uncertainty differently depending on condition (partial or complete feedback) and/or choice horizon (long and short). We thus constructed contrasts adding up and subtracting the first choices on long and short horizons for the constant, the expected value, and the uncertainty. At outcome, we were interested in reward effects and updates to the expected value of stimuli. As this should happen not just after first choices in a horizon but all choices, we used contrast to construct (weighted) averages of our regressors combining all choices within horizons. Moreover, to look at the effect of (signed) prediction errors, we use contrasts that subtract the expectation from the reward.

To visualize the cluster-corrected effects we find in our mask of the frontal cortex, we use an atlas of the macaque brain [36] to identify the regions where we observe activity. We then create ROIs by calculating the overlap of the anatomical region according to the atlas (dilated with a kernel of $3 \times 3 \times 3$ voxels), and the functional activation we found. By extracting the average t-statistics in this region, we are able to visualize the effects we found and also examine the individual components that contributed the effects (e.g., the reward and outcome expectation for prediction errors).

To best describe the localization of orbitofrontal activities, we considered 3 orbital subdivisions based on their respective position on the orbital surface. Lateral to the lateral orbitofrontal sulcus is the lateral OFC; medial to the medial orbitofrontal sulcus is the medial OFC. In between the two sulci is a region we referred to as the central OFC. Such parcellation resembles subdivisions considered in humans and rodents [71,85,86], although alternative labels have been proposed [70].

To best describe the localization of cingulate activities, we considered a dissociation between anterior and mid-cingulate subdivisions as proposed by Vogt and colleagues [87,88].

## Supporting information

**S1 Fig. Full model fit of the model predicting choosing the right option on screen on first choices (shown in Fig 2D and described in detail in the Materials and methods section).** (**A**) Predictors are from left to right: Intercept (i.e., a side bias), repetition bias (RB), expected value of difference between right and left according to our Bayesian model (EV), uncertainty difference between right and left according to our Bayesian model (U), horizon length (short horizon is positive, long horizon is negative), the interaction between horizon and expected value (horizonXER), and the interaction between horizon and uncertainty (horizonXU). The distributions are the posteriors of the parameter estimates, shown both for each monkey individually and averaged over animals. Fits from the partial feedback sessions are shown on the left, and from the complete feedback sessions on the right. (**B**) Data from the same fit as in (A) but now summed up over both partial and complete feedback sessions. (**C**) Data from the same fit as in (A) but now we computed the difference between partial and complete feedback sessions. (**D**) One-sided p-values for all parameters are computed as the number of samples of the posterior greater than 0. To compute the p-value for effects smaller than 0, the p-values in the table can be subtracted from 1. Data and code to reproduce the figure can be found at https://doi.org/10.5281/zenodo.7464572.
(TIFF)

**S2 Fig. The same model as in S1 Fig but only fit to trials during which the available choices on screen were the same on each side (2 and 2).** All conventions are the same as in S1 Fig. Data and code to reproduce the figure can be found at https://doi.org/10.5281/zenodo.7464572.
(TIFF)

**S3 Fig. Equivalency between our linear regression and the framing in terms of *random* and *directed* exploration.** (**A**) With the uncertainty regressor. We find that monkeys modulate their sensitivity to the expected value depending on the horizon and the feedback type, which is equivalent to the *random* exploration parameter, the softmax noise, which is the inverse of the expected value regressor. However, we find no modulation of the uncertainty by the horizon nor the feedback type, which is equivalent to the *directed* exploration parameter, the uncertainty bonus, which is the uncertainty regressor divided by the expected value regressor. (**B**) Same as A but with the number of available information rather than the uncertainty. Error bars indicate standard deviation. Data and code to reproduce the figure can be found at https://doi.org/10.5281/zenodo.7464572. (TIFF)

**S4 Fig. The same model as in S1 Fig but only using the number of available information on each side rather than the uncertainty.** All conventions are the same as in S1 Fig. Data and code to reproduce the figure can be found at https://doi.org/10.5281/zenodo.7464572. (TIFF)

**S5 Fig. Full model fit of predicting choosing the right option on screen during subsequent choices in the long horizon (choices 2–4; shown in Fig 3E and described in detail in the Materials and methods section).** (**A**) Predictors are from left to right: Intercept (i.e., a side bias), repetition bias (RB), the change in expected value between the right and left option revealed by choices made during this horizon, compared to the initial expected value for this horizon, i.e., the baseline (deltaERchosen), the change in expected value between the right and left option revealed by feedback about the unchosen option, compared to the initial expected value for this horizon (deltaERcounterfactual), the difference in initial expected value between the right and left option available, i.e., the expected value difference at first choice (baselineU), the change in uncertainty between the right and left option revealed by choices made during this horizon, compared to the initial uncertainty for this horizon (deltaUchosen), the change in uncertainty between the right and left option revealed by feedback about the unchosen option, compared to the initial uncertainty for this horizon (deltaUcounterfactual), the difference in initial uncertainty between the right and left option available, i.e., the uncertainty difference at first choice (baselineU), the difference between how often the right option has been chosen over the left option during this horizon (deltaChosen). All other conventions are the same as in S1 Fig, also for panels B-D. Data and code to reproduce the figure can be found at https://doi.org/10.5281/zenodo.7464572. (TIFF)

**S6 Fig.** (**A**) Expected value of the chosen option without mask and when taking the activity before the choice in all trials (not just first choice trials), we observed large activations related to the expected value of the chosen option (which is the same as the chosen action in our task) spanning from the motor cortex/somatosensory cortex, the dlPFC, the OFC, and striatum, as well as an inverted signal in the visual areas (Cluster $p < 0.05$, cluster forming threshold of $z > 2.3$). (**B**) In the partial and complete feedback conditions in our VOI and when taking the activity before the first choice only, we found 1 cluster of activity related to the inverse of the magnitude of the uncertainty about the chosen option in the right medial prefrontal cortex (24c and 9m) that extended bilaterally in the frontal pole (10mr) (Cluster $P < 0.05$, cluster forming threshold of $z > 2.3$). Data to reproduce the figure can be found at https://doi.org/10.5281/zenodo.7464572. dlPFC, dorsolateral prefrontal cortex; OFC, orbitofrontal cortex; VOI, volume of interest. (TIFF)

**S7 Fig. Outcome prediction error and magnitude in the partial feedback condition.** (**A**) In the partial feedback condition and at the time of outcome, we found 3 clusters of activity that were positively modulated by the chosen option prediction error in the medial prefrontal cortex and bilaterally in the somatosensory and motor cortex in our VOI (Cluster $p < 0.05$, cluster forming threshold of $z > 2.3$). (**B**) We found the same 3 clusters when we looked for a positive modulation by the magnitude of the chosen outcome. We additionally found 1 cluster of activity in the right lateral OFC that was negatively modulated by the magnitude of the chosen outcome. (**C**) When we time-locked our search to the onset of the reward (1 s after the display of the outcome, with a different GLM), we found the same clusters as in A, as well as the classic prediction error related activity in the ventral striatum and a negative prediction error in visual areas (see full map at https://doi.org/10.5281/zenodo.7464572) at the whole brain level. Data to reproduce the figure can be found at https://doi.org/10.5281/zenodo.7464572. GLM, general linear model; OFC, orbitofrontal cortex; VOI, volume of interest.
(TIFF)

**S8 Fig. Chosen and unchosen outcome magnitude in the complete feedback condition.** (**A**) In complete feedback sessions only, we found clusters for inverted chosen outcome magnitude activity in the right lOFC (47/12o) and bilaterally in the vlPFC and 2 clusters in the somatosensory/motor cortex [3]. (**B**) We found a cluster of activity for the inverted unchosen outcome magnitude in the cOFC and mOFC and the vlPFC. Data to reproduce the figure can be found at https://doi.org/10.5281/zenodo.7464572. cOFC, central orbitofrontal cortex; lOFC, lateral orbitofrontal cortex; mOFC, medial orbitofrontal cortex; vlPFC, ventrolateral prefrontal cortex.
(TIFF)

**S1 Table. Tables showing the peaks of all significant clusters found within our frontal masks that are reported in the main text.** Coordinates are given in the F99 standard space.
(TIFF)

## Acknowledgments

We thank Drs. Kevin Marche, Lea Roumazeilles, and Urs Schuffelgen, as well as Kelly Simpson for technical assistance during the data acquisition and the animal housing facility staff for their care of the animals. We thank the Motivation Brain and Behavior lab, the Rushworth lab, the Walton lab, Dr. Nadescha Trudel, and Dr. Vasilisa Skvortsova for insightful conversations that helped shape this manuscript.

## Author Contributions

**Conceptualization:** Caroline I. Jahn, Jan Grohn, Sebastien Bouret, Mark E. Walton, Nils Kolling, Jérôme Sallet.

**Formal analysis:** Caroline I. Jahn, Jan Grohn.

**Investigation:** Caroline I. Jahn, Jan Grohn, Steven Cuell, Andrew Emberton, Jérôme Sallet.

**Methodology:** Caroline I. Jahn, Jan Grohn, Nils Kolling, Jérôme Sallet.

**Visualization:** Caroline I. Jahn, Jan Grohn.

**Writing – original draft:** Caroline I. Jahn, Jan Grohn, Nils Kolling, Jérôme Sallet.

**Writing – review & editing:** Caroline I. Jahn, Jan Grohn, Steven Cuell, Sebastien Bouret, Mark E. Walton.

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
