## [Editor Report · Decision Letter 0]

27 Jun 2022

Dear Dr Jahn, 

Thank you for submitting your manuscript entitled "Strategic exploration in the macaque’s prefrontal cortex." for consideration as a Research Article by PLOS Biology.

Your manuscript has now been evaluated by the PLOS Biology editorial staff, as well as by an academic editor with relevant expertise, and I am writing to let you know that we would like to send your submission out for external peer review.

Once your full submission is complete, your paper will undergo a series of checks in preparation for peer review. After your manuscript has passed the checks it will be sent out for review. To provide the metadata for your submission, please Login to Editorial Manager (https://www.editorialmanager.com/pbiology) within two working days, i.e. by Jun 29 2022 11:59PM.

Kind regards,

Kris

Kris Dickson, Ph.D. (she/her)

Neurosciences Senior Editor/Section Manager

PLOS Biology

kdickson@plos.org

---

## [Decision Letter · Decision Letter 1]

14 Aug 2022

Dear Dr Jahn,

Thank you for your patience while your manuscript "Strategic exploration in the macaque’s prefrontal cortex." was peer-reviewed at PLOS Biology. It has now been evaluated by the PLOS Biology editors, an Academic Editor with relevant expertise, and by several independent reviewers. 

In light of the reviews, which you will find at the end of this email, we would like to invite you to revise the work to thoroughly and comprehensively address the reviewers' reports.

Given the extent of revision needed, we cannot make a decision about publication until we have seen the revised manuscript and your response to the reviewers' comments. Your revised manuscript is likely to be sent for further evaluation by all or a subset of the reviewers.

We hope to receive your revised manuscript within 3 months. If you feel you might need additional time to comprehensively address the reviewer concerns or you have additional questions, please email us at plosbiology@plos.org. 

**IMPORTANT - SUBMITTING YOUR REVISION**

*Re-submission Checklist*

*Published Peer Review*

*PLOS Data Policy*

*Blot and Gel Data Policy*

Sincerely,

Kris

Kris Dickson, Ph.D. (she/her)

Neurosciences Senior Editor/Section Manager

PLOS Biology

kdickson@plos.org

REVIEWS:

Reviewer #1: In this new interesting work, Jahn and colleagues studied the behavioral and neural correlates of strategic decision-making in macaques. Through an innovative adaptation of the horizon task, previously used in human research by another group of researchers, the authors were able to study how monkeys strategically made different, well-defined, choices constrained by short vs. long decision horizon manipulations. Then, the authors used this task to examine the neural (fMRI) correlates of these behaviors.

While behavioral and neural bases of exploration vs. exploitation strategic decisions in rhesus macaques have been studied frequently through multiple approaches in the past, the use of the horizon task provided a novel and unique context to examine these strategic behaviors. In particular, this study could rely on the 'functional utility' of exploration decisions in a more constrained manner than some of the other existing studies due to the need to continue choosing sequentially in the long horizon condition. Furthermore, the use of fMRI allowed more holistic perspectives on how different decision variables are represented in this task. 

Overall, the use of a novel task design combined with the careful analyses of decision variables at both behavioral and neural levels makes this research influential for future research in the domain of strategic decision-making. The behavioral and neural results are well-grounded in the model used and reveal an important change in expected value representation to promote exploration when there is a strategic benefit for doing so. 

I believe this is a strong study with very interesting results. I did not find any major issues with this work but have some (mostly minor) comments that could further improve the manuscript and enhance its broader impact.

1. A rather big portion of the Introduction is taken up by describing the task logic, methods, and results of the current work. Much of this information is again repeated at the beginning of the Results section. I suggest the authors cover more literature background and research motivation in the Introduction beyond introducing the horizon task and its relation to what the authors did in this manuscript. Doing so will more strongly position this work in light of the big-picture importance of studying strategic decision-making, in the brain and help readers, in addition to the Discussion section, to understand why this work is aimed at providing novel insights into strategic decision-making than other existing studies.

2. Strategic decisions frequently have idiosyncratic components as different monkeys may employ slightly different strategies for solving the same task even when their overall strategies might converge on average. It might be thus worthwhile to examine if there were individual differences in strategies over the course of a session or the course of the entire data collection (e.g., were these decision variables stationary or non-stationary?). Given that there is evidence of individual differences in Supp Figures 1-3, it would be informative to go deeper into how such individual differences might be reflected potentially differently across ACC/MCC, dlPFC, and OFC BOLD activations with respect to their functional roles suggested in this manuscript.

3. In P. 9, "However, we found no statistically reliable difference in the sensitivity to the uncertainty across the experimental conditions. Therefore, uncertainty did not play a key role in strategic exploration in our task." I am a bit surprised that there were no clear effects of uncertainty in monkeys' behaviors, as reducing uncertainty should be a major functional driver of decisions that are central to exploration. Unless I missed it, I don't see a place where the authors examined the effects of uncertainty at the neural activation level and how it might have influenced the neural representation of strategic decision variables in the brain regions examined. It would be surprising if uncertainty (variance) played no role in ACC/MCC, dlPFC, or OFC activations on signaling expected value or chosen/unchosen outcomes. If there are any effects, these modulations might further provide neural mechanistic insights into the computations performed by these brain regions for explore vs. exploit decisions. Either way, I suggest the authors discuss in some detail the effects of uncertainty (behavioral and neural) in the Discussion in relation to the main findings.

4. Throughout the manuscript, while the innovative and purposeful use of the horizon task was well discussed, the big-picture importance of having the functional utility of exploring vs. exploiting in the horizon condition compared to other existing studies of explore vs. exploit decision paradigms (e.g., two-armed bandit with fixed probability or with dynamic reward schedule) was not communicated as effectively as possible. Perhaps the authors should make this point clearly so that the readers can better grasp why this particular study was able to demonstrate certain aspects of behavioral and neural strategic decision-making in monkeys compared to other existing work.

5. Differentiated or common roles of different brain regions tracking decision variables were interesting. Were there any different functional connectivity patterns among different brain regions for exploration vs. exploitation and in long vs. short horizons? Such network-level analyses may provide additional support on how different brain regions work together to compute strategic choices.

6. Fig. 1B, in the distributions of Expected Value and Uncertainty, there are some thin lines that should be deleted (second and third column of the distribution graphs) - I believe these are remainders left over while making the figure.

7. P. 11, the results section at the top of this page has some sentences that should refer to Fig. 3E, rather than Fig. 3D.

Reviewer #2: Jahn CI et al., Strategic exploration in the macaque's prefrontal cortex

The authors describe data from a study in which they trained monkeys on a primate variation of the horizon's task, to study the explore-exploit tradeoff, and carried out fMRI while animals executed the task. They found that animals relied more on the expected value in short horizon than long horizon choices. There choices did not, however, depend on the uncertainty of the options. Comparisons of different conditions showed that there was increased activation in areas 24, 46 and 47/12 for long-horizon vs. short-horizon trials, and increased activation in area 46 for partial vs. complete feedback. 

This manuscript addresses important questions about the behavioral and neural mechanisms that underlie exploration vs. exploitation. The behavioral paradigm is well-controlled and is based on the Horizon's task developed previously by Robert Wilson. This task allows for explicitly testing specific hypotheses about the factors that underlie exploratory behavior. Overall, this is an interesting manuscript. I do have several comments which should be addressed. 

Comments

1. The abstract and introduction conflate foraging and the explore-exploit tradeoff. Foraging and the explore-exploit tradeoff, however, refer to different behavioral/theoretical processes. Stephens and Krebs (reference 1) developed the marginal value theorem to account for foraging. Within this framework, the choice to leave a patch results in a random sample from a known distribution of other patches. Thus, there is no learning within the theoretical framework that has been developed to describe foraging. The explore-exploit tradeoff on the other hand comes from the reinforcement learning literature and refers specifically to the process being studied in this manuscript. In the explore-exploit tradeoff the goal is specifically to learn, or gather more information, about some probability distribution. While there is of course an intuitive or folk similarity in the terms exploration and forage, they do not refer to the same phenomenon as studied in the literature. If you would like to stick with the term foraging, I would suggest stating that you are not using it in the way in which it was originally defined. I would, however, suggest removing reference to foraging.

2. A number of papers, some of them recent, should be cited. The work of Vincent Costa in monkeys and recently in humans (Hogeveen J et al., Neuron, 2022; Costa VD and Averbeck, J Neurosci, 2020; Costa VD et al., Neuron, 2019). Work by Becket Ebitz (Ebitz RB et al., Neuron, 2018), and work by Michael Frank (Badre D et al., Neuron, 2012; Cavanagh JF et al., Cer Cortex, 2012), for example. 

3. It would be useful to report ANOVA results for the data shown in Fig. 2B. A direct examination of main and interaction effects of feedback and horizon would be useful. Means from each session could be entered into the analysis. It's also somewhat surprising that the regression weight for expected value is higher in the complete feedback condition (Fig. 2D) but the probability of choosing the highest EV option is not higher in 2B. Perhaps some of the other variables from the logistic regression are driving this. Are there strong correlations, for example, between repetition bias and EV?

4. In the Horizon's task, exploration is divided into directed and undirected (or random) exploration. In the current manuscript, the term strategic exploration is used. However, there are no effects of uncertainty on exploration, and therefore monkeys are only using undirected exploration. Is random exploration strategic? Is the strategic referring to the increase in exploration in the partial feedback and long horizon in the partial condition? Do these effects come out of the ANOVA (comment 3?). It would be useful to discuss explicitly, directed and undirect exploration, and how these relate to the current results. Avoiding uncertainty, which the animals show, is difficult to fit within this framework. Perhaps some mention of this would be useful. 

5. The figure legends state that error bars are S.E.M. but they do not give the N. 

6. F and t- stats and N are given for the imaging data, but not for the behavioral data. More detailed statistics should be given for the behavioral data.

7. The Bayesian statistics are fine. But what if a t-test were used on the parameter comparisons for the logistic regression? Can you report t and p values for these comparisons? It's not clear what is gained by using the Bayesian framework in this situation. 

Reviewer #3: This manuscript describes an attempt to implement the horizon task pioneered by Wilson and colleagues (2013) in rhesus macaques. An operating assumption is that monkeys like humans would use directed exploration, guided by uncertainty, in deciding whether to sample one option over another. To test this assumption authors adapted the version of the task tested in humans. The monkeys first observed visual cue presentations that signaled information about the potential value in choosing either cue. Equal (2 observations of visual feedback about each cue) or unequal information (3 observations of visual feedback for one cue and 1 observation of the other cue) was provided about each of the two cues during this observation phase. No primary rewards were experienced during the observation phase. The monkeys were then visually signaled by the height of two gray vertical bars on the left and right side of the screen, that they would have either one (short horizon) or four (long horizon) opportunities to choose between the two options following the observation phase. After each choice, the monkeys received between 0 and 10 drops of juice determined by the length of visual feedback associated with their choice. The monkeys completed the task inside an MRI scanner to enable assessments of brain activity related to encoding of the choice horizon and expected value. One additional manipulation was that following their choices the monkeys were provided with either partial feedback (i.e. they only received visual feedback about the option they chose) or complete feedback (i.e. they were shown visual feedback about both the chosen and unchosen option, but only received primary reward related to the option they chose). The partial versus complete feedback manipulation occurred across the individual test sessions, whereas the horizon and information manipulations occurred within a session. The authors used a Bayesian model to quantify the expected value and uncertainty associated with choosing each option prior to each instance of visual feedback in the observation and choice phases of each trial. The authors focus on a reduced reliance on the expected value in deciding which option to choose as indicative of the monkeys' using "strategic exploration" to decide when to explore, predicting increased exploration in long versus short horizons when the monkeys received partial feedback. Whereas exploration should not differ between the horizon conditions when complete feedback is provided. Although the authors believe they have sufficient behavior and neural evidence to support this hypothesis, their claims are not well supported. I have serious doubts about whether the monkeys were even sensitive to the horizon manipulation and more fundamental behavioral analyses that hew closely to the data and match those performed previously in humans are needed before interpreting the reported fMRI contrasts or claiming to have identified differences in humans versus non-human primates use of strategic exploration. My major concerns are outlined below:

1. The authors need to provide more direct evidence, replicating analyses from Wilson et al. (2013) that demonstrates the monkeys were sensitive to the horizon manipulation. While their accuracy is above chance it is not apparent that the accuracy measures directly differ by the horizon or feedback conditions based on the error bars in the plot provided in Figure 2. The horizon task is set up to measure directed versus random exploration, explicitly, by assessing how either the selection of the more informative, uncertain option varies as a function of the difference in the expected value of the two options based on the observation phase when unequal information is provided (i.e. the number of forced choice trials in the observation) or preference for a particular choice option (i.e. right or left) varies with the difference in the expected value when equal information is provided. When a non-linear choice function (e.g. sigmoid) is fit to the behavior in this way the slope and intercept of the function can be used to quantify random exploration (slope) and how directed exploration is shaped by an information bonus. Given that the manuscript potentially describes the first implementation of the horizon task in nonhuman primates, these fundamental analyses that build a bridge to human studies and that do not rely on the implementation of a particular computational model must be added to the main body of the manuscript. Especially critical is fitting these functions to choices on the first trial in the partial feedback sessions when equal or unequal information is presented. But I suspect, given the result from the Bayesian model that the monkeys avoided uncertainty, that these analyses will reveal that the monkeys are not using directed exploration at all, that the information bonuses are equivalent across the two horizons, and that the documented "strategic exploration" is simply an increase in random exploration denoted by a smaller slope in the long vs. short horizon condition stemming from the animals being less exploitative. If this is the case then it casts the fMRI analyses in a very different light, especially the contrast of the activity as a function of horizon. If this contrast is effectively highlighting random rather than directed exploration this might be one reason that cingulate rather than the frontal pole of prefrontal cortex is activated differently under the two conditions. To summarize, the authors need to show, using some behavioral metric that doesn't rely on the Bayesian regression, using only the first choice trials where value and uncertainty are decoupled by the forced choice manipulation that convincingly shows the animals were sensitive to the horizon manipulation. 

2. The partial and complete feedback condition comparisons is problematic in isolating processing of counterfactual feedback. This is because in the complete feedback sessions where the feedback is always provided during the forced and free choice trials for both options, the counterfactual feedback is confounded with complete (i.e. equal) information, whereas in the partial feedback sessions the amount of information provided is either equal or unequal. Instead of a direct comparison between the partial and feedback sessions, the authors should be contrasting a randomly sampled subset of trials from the complete feedback sessions with trials in which equal amounts of partial feedback were provided about the two cues in the observation phase. Ideally, the same exact feedback would be provided but I recognize this might be impossible. This contrast and the identified neural activity could then be compared to the currently missing contrast of equal and unequal feedback in the partial feedback sessions, to better isolate the effects due to counterfactual feedback versus uncertainty. 

3. The use of one-tailed statistical tests is not defensible and many of the key effects, particularly relating to the horizon manipulations would not survive appropriate corrections for multiple comparisons.

---

## [Decision Letter · Decision Letter 2]

16 Dec 2022

Dear Dr Jahn,

Thank you for your patience while we considered your revised manuscript "Strategic exploration in the macaque’s prefrontal cortex." for publication as a Research Article at PLOS Biology. This revised version of your manuscript has been evaluated by the PLOS Biology editors, the Academic Editor and the original reviewers.

Based on the reviews and on our Academic Editor's assessment of your revision, we are likely to accept this manuscript for publication. Given that our readership might have similar questions to those raised by the reviewers, we would like you to do some rewriting of your manuscript to incorporate the point-by-point responses you'd provided to Reviewer 1 into the body of your manuscript. We also ask that you consider a title change to something like:

The macaque prefrontal cortex supports strategic exploration (If you don't feel this is too strongly worded)

OR

Neural responses in macaque prefrontal cortex are linked to strategic exploration

Please also make sure to fully address the data and other policy-related requests at the bottom of this email. Failure to do so will result in delays in moving your submission forward.

We expect to receive your revised manuscript within two weeks. 

*Published Peer Review History*

*Press*

Sincerely,

Kris

Kris Dickson, Ph.D., (she/her)

Neurosciences Senior Editor/Section Manager,

kdickson@plos.org,

PLOS Biology

ETHICS STATEMENT:

Please update your ethics statement with the required additional details as outlined in

https://journals.plos.org/plosbiology/s/animal-research#loc-non-human-primates

Specifically, non-human primate studies must be performed in accordance with the recommendations of the Weatherall report “The use of non-human primates in research”. Manuscripts describing research involving non-human primates must include details of animal welfare, including information about housing, feeding, and environmental enrichment, and steps taken to minimize suffering, including use of anesthesia and method of sacrifice if appropriate.

DATA POLICY:

We appreciate the deposition of your data on Gitfront but note that we cannot accept sole deposition of data to non-static site (e.g. no personal sites and generally no institutional sites). (https://journals.plos.org/plosbiology/s/data-availability). We require deposition of all summary data to a static site, like Zenodo, FigShare or OSF. GitHub and similar sites can be used for depositing code however.

Note that we do not require all raw data to be deposited on such a static site. Rather, we ask that all individual quantitative observations that underlie the data summarized in the figures and results of your paper be made available in order to allow our readership to reproduce the figures in your paper:

1) Please ensure that these data files are invariably referred to (in the manuscript, figure legends, and the Description field when uploading your files) using the following format verbatim: S1 Data, S2 Data, etc. If using excel, multiple panels of a single or even several figures can be included as multiple sheets in one excel file that is saved using exactly the following convention: S1_Data.xlsx (using an underscore).

2) Please also provide the updated accession code so that we may view your data before publication. 

3) Please ensure that you provide the individual numerical values that underlie the summary data displayed in the following figure panels as they are essential for readers to assess your analysis and to reproduce it:

Fig2A-D; Fig3A-E; Fig4B,D,E; Fig5B-D

Supplemental figures: S1A,B; S2A-C; S3A,B; S4A-C; S5A-C;

4) Please also ensure that figure legends in your manuscript include information on where this underlying data can be found (e.g. “The underlying data supporting Fig X, panel Y can be found in file Z.”), and please also ensure that your supplemental data file/s has a legend.

DATA NOT SHOWN?

- Please note that per journal policy, we do not allow the mention of "data not shown", "personal communication", "manuscript in preparation" or other references to data that is not publicly available or contained within this manuscript. Please check your submission to ensure no such statements are included. If there are such statements, either remove mention of these data or provide figures presenting the results and the data underlying the figure(s).

Reviewer remarks:

Do you want your identity to be public for this peer review?

Reviewer #1: Yes: Steve W. C. Chang

Reviewer #2: No

Reviewer #3: No

Reviewer #1: The authors did a very thorough job at addressing all of my points by 1) performing a number of new analyses to support their responses, 2) adding new information on the manuscript, and 3) modifying their interpretations more broadly in the context of explore vs. exploit decision-making. Their responses were detailed and constructed based on evidence from their data (even though not included in the manuscript). I have no more comments.

Reviewer #2: The authors have addressed my concerns. I have no further comments.

Reviewer #3: The authors have have satisfactorily addressed my previous concerns.

---

## [Editor Report · Decision Letter 3]

3 Jan 2023

Dear Dr Jahn,

Thank you for the submission of your revised Research Article "Neural responses in macaque prefrontal cortex are linked to strategic exploration" for publication in PLOS Biology. On behalf of my colleagues and the Academic Editor, Thorsten Kahnt, I am pleased to say that we can in principle accept your manuscript for publication. In your final version, we do ask that you either include the two additional statements addressing Dr Chang's points (i.e. "Based on the reviewer's request...") and include a statement at the end of these paragraphs referring readers to the supplemental "response to reviewers" file that you will also include with the final submission, or minimally that you include a pared down version of these two statements: "We conducted additional exploratory brain-behavior correlations but found no significant relationships to behavioral sensitivity (see supplemental "Response to Reviewers" file for additional details)." Please also address any remaining formatting and reporting issues that will be detailed in an email you should receive within 2-3 business days from our colleagues in the journal operations team; no action is required from you until then. Please note that we will not be able to formally accept your manuscript and schedule it for publication until you have completed these requested changes.

PRESS

We frequently collaborate with press offices. If your institution or institutions have a press office, please notify them about your upcoming paper at this point, to enable them to help maximize its impact. If the press office is planning to promote your findings, we would be grateful if they could coordinate with biologypress@plos.org. If you have previously opted in to the early version process, we ask that you notify us immediately of any press plans so that we may opt out on your behalf.

With best wishes for 2023, 

Kris

Kris Dickson, Ph.D., (she/her)

Neurosciences Senior Editor/Section Manager

PLOS Biology

kdickson@plos.org